# Adversarial Intrinsic Motivation for Reinforcement Learning

**Ishan Durugkar**
Department of Computer Science
The University of Texas at Austin
Austin, TX, USA 78703
ishand@cs.utexas.edu

**Mauricio Tec**
Department of Statistics and Data Sciences
The University of Texas at Austin
Austin, TX, USA 78703
mauriciogtec@utexas.edu

**Scott Niekum**
Department of Computer Science
The University of Texas at Austin
Austin, TX, USA 78703
sniekum@cs.utexas.edu

**Peter Stone**
Department of Computer Science
The University of Texas at Austin
Austin, TX, USA 78703 and
Sony AI
pstone@cs.utexas.edu

## Abstract

Learning with an objective to minimize the mismatch with a reference distribution has been shown to be useful for generative modeling and imitation learning. In this paper, we investigate whether one such objective, the Wasserstein-1 distance between a policy's state visitation distribution and a target distribution, can be utilized effectively for reinforcement learning (RL) tasks. Specifically, this paper focuses on goal-conditioned reinforcement learning where the idealized (unachievable) target distribution has full measure at the goal. This paper introduces a quasimetric specific to Markov Decision Processes (MDPs) and uses this quasimetric to estimate the above Wasserstein-1 distance. It further shows that the policy that minimizes this Wasserstein-1 distance is the policy that reaches the goal in as few steps as possible. Our approach, termed Adversarial Intrinsic Motivation (AIM), estimates this Wasserstein-1 distance through its dual objective and uses it to compute a supplemental reward function. Our experiments show that this reward function changes smoothly with respect to transitions in the MDP and directs the agent's exploration to find the goal efficiently. Additionally, we combine AIM with Hindsight Experience Replay (HER) and show that the resulting algorithm accelerates learning significantly on several simulated robotics tasks when compared to other rewards that encourage exploration or accelerate learning.

## 1 Introduction

Reinforcement Learning (RL) [74] deals with the problem of learning a policy to accomplish a given task in an optimal manner. This task is typically communicated to the agent by means of a reward function. If the reward function is sparse [4] (e.g., most transitions yield a reward of 0), much random exploration might be needed before the agent experiences any signal relevant to learning [11, 2].

Some of the different ways to speed up reinforcement learning by modifying or augmenting the reward function are shaped rewards [52], redistributed rewards [2], intrinsic motivations [8, 69, 71, 72, 54, 57], and learned rewards [81, 54]. Unfortunately, the optimal policy under such modified rewards might sometimes be different than the optimal policy under the task reward [52, 18]. The problem of

35th Conference on Neural Information Processing Systems (NeurIPS 2021).

learning a reward signal that speeds up learning by communicating *what to do* but does not interfere by specifying *how to do it* is thus a useful and complex one [82].

This work considers whether a task-dependent reward function learned based on the distribution mismatch between the agent's state visitation distribution and a target task (expressed as a distribution) can guide the agent towards accomplishing this task. Adversarial methods to minimize distribution mismatch have been used with great success in generative modeling [29] and imitation learning [39, 25, 79, 76, 28]. In both these scenarios, the task is generally to minimize the mismatch with a target distribution induced by the data or expert demonstrations. Instead, we consider the task of goal-conditioned RL, where the ideal target distribution assigns full measure to a goal state. While the agent can never match this idealized target distribution perfectly unless starting at the goal, intuitively, minimizing the mismatch with this distribution should lead to trajectories that maximize the proportion of the time spent at the goal, thereby prioritizing transitions essential to doing so.

The theory of optimal transport [78] gives us a way to measure the distance between two distributions (called the Wasserstein distance) even if they have disjoint support. Previous work [3, 32] has shown how a neural network approximating a potential function may be used to estimate the Wasserstein-1 distance using its dual formulation, but assumes that the metric space this distance is calculated on is Euclidean. A Euclidean metric might not be the appropriate metric to use in more general RL tasks however, such as navigating in a maze or environments where the state features change sharply with transitions in the environment.

This paper introduces a quasimetric tailored to Markov Decision Processes (MDPs), the time-step metric, to measure the Wasserstein distance between the agent's state visitation distribution and the idealized target distribution. While this time-step metric could be an informative reward on its own, estimating it is a problem as hard as policy evaluation [31]. Instead, we show that the dual objective, which maximizes difference in potentials while utilizing the structure of this quasimetric for the necessary regularization, can be optimized through sampled transitions.

We use this dual objective to estimate the Wasserstein-1 distance and propose a reward function based on this estimated distance. An agent that maximizes returns under this reward minimizes this Wasserstein-1 distance. The competing objectives of maximizing the difference in potentials for estimating the Wasserstein distance and minimizing it through reinforcement learning on the subsequent reward function leads to our algorithm, Adversarial Intrinsic Motivation (AIM).

Our analysis shows that if the above Wasserstein-1 distance is computed using the time-step metric, then minimizing it leads to a policy that reaches the goal in the minimum expected number of steps. It also shows that if the environment dynamics are deterministic, then this policy is the optimal policy.

In practice, minimizing the Wasserstein distance works well even when the environment dynamics are stochastic. Our experiments show that AIM learns a reward function that changes smoothly with transitions in the environment. We further conduct experiments on the family of goal-conditioned reinforcement learning problems [1, 63] and show that AIM when used along with hindsight experience replay (HER) greatly accelerates learning of an effective goal-conditioned policy compared to learning with HER and the sparse task reward. Further, our experiments show that this acceleration is similar to the acceleration observed by using the actual distance to the goal as a dense reward.

## 2   Related Work

We highlight the related work based on the various aspects of learning that this work touches, namely intrinsic motivation, goal-conditioned reinforcement learning, and adversarial imitation learning.

### 2.1   Intrinsic Motivation

Intrinsic motivations [8, 57, 56] are rewards presented by an agent to itself in addition to the external task-specific reward. Researchers have pointed out that such intrinsic motivations are a characteristic of naturally intelligent and curious agents [30, 5, 6]. Intrinsic motivation has been proposed as a way to encourage RL agents to learn skills [10, 9, 68, 62] that might be useful across a variety of tasks, or as a way to encourage exploration [11, 67, 7, 24, 23]. The techniques that encourage general exploration [16, 58] have also been studied as inducing an artifical curiosity [66, 65]. The optimal reward framework [69, 72] and shaped rewards [52] (if generated by the agent itself) are a way to

assist an RL agent in learning the optimal policy for a given task. Such an intrinsically motivated reward signal has previously been learned through various methods such as evolutionary techniques [54, 64], meta-gradient approaches [71, 81, 82], and others. The Wasserstein distance has been used to present a valid reward for imitation learning [79, 19] as well as program synthesis [27].

## 2.2 Goal-Conditioned Reinforcement Learning

Goal-conditioned reinforcement learning [43] can be considered a form of multi-task reinforcement learning [17] where the agent is given the goal state it needs to reach at the beginning of every episode, and the reward function is sparse with a non-zero reward only on reaching the goal state. UVFA [63], HER [1], and others [80, 20] consider this problem of reaching certain states in the environment. Relevant to our work, Venkattaramanujam et al. [77] learns a distance between states using a random walk that is then used to shape rewards and speed up learning, but requires goals to be visited before the distance estimate is useful. DisCo RL [51] extends the idea of goal-conditioned RL to distribution-conditioned RL.

Contemporaneously, Eysenbach et al. [21, 22] has proposed a method which considers goals and examples of success and tries to predict and maximize the likelihood of seeing those examples under the current policy and trajectory. For successful training, this approach needs the agent to actually experience the goals or successes. Their solution minimizes the Hellinger distance to the goal, a form of $f$-divergence. AIM instead uses the Wasserstein distance which is theoretically more informative when considering distributions that are disjoint, and does not require the assumption that the agent has already reached the goal through random exploration. Our experiments in fact verify the hypothesis that AIM induces a form of directed exploration in order to reach the goal.

## 2.3 Adversarial Imitation Learning and Minimizing Distribution Mismatch

Adversarial imitation learning [39, 25, 79, 76, 28] has been shown to be an effective method to learn agent policies that minimize distribution mismatch between an agent's state-action visitation distribution and the state-action visitation distribution induced by an expert's trajectories. In most cases this distribution that the expert induces is achievable by the agent and hence these techniques aim to match the expert distribution exactly. In the context of goal-conditioned reinforcement learning, GoalGAIL [20] uses adversarial imitation learning with a few expert demonstrations to accelerate the learning of a goal-conditioned policy. In this work, we focus on unrealizable target distributions that cannot be completely matched by the agent, and indeed, are not induced by any trajectory distribution.

FAIRL [28] is an adversarial imitation learning technique which minimizes the Forward KL divergence and has been shown experimentally to cover some hand-specified state distributions, given a smoothness regularization as used by WGAN [32]. $f$-IRL [53] learns a reward function where the optimal policy matches the expert distribution under the more general family of $f$-divergences. Further, techniques beyond imitation learning [46, 37] have looked at matching a uniform distribution over states to guarantee efficient exploration.

# 3 Background

In this section we first set up the goal-conditioned reinforcement learning problem, and then give a brief overview of optimal transport.

## 3.1 Goal-Conditioned Reinforcement Learning

Consider a goal-conditioned MDP as the tuple $\langle \mathcal{S}, \mathcal{A}, \mathcal{G}, P, \rho_0, \sigma, \gamma \rangle$ with discrete state space $\mathcal{S}$, discrete action space $\mathcal{A}$, a subset of states which is the goal set $\mathcal{G} \subseteq \mathcal{S}$, and transition dynamics $P : \mathcal{S} \times \mathcal{A} \times \mathcal{G} \longmapsto \Delta(\mathcal{S})$ ($\Delta(\cdot)$ is a distribution over a set) which might vary based on the goal (see below). $\rho_0 : \Delta(\mathcal{S})$ is the starting state distribution, and $\sigma : \Delta(\mathcal{G})$ is the distribution a goal is drawn from. $\gamma \in [0, 1)$ is the discount factor. We use discrete states and actions for ease of exposition, but our idea extends to continuous states and actions, as seen in the experiments.

At the beginning of an episode, the starting state is drawn from $\rho_0$ and the goal for that episode is drawn from $\sigma$. The reward function $r : \mathcal{S} \times \mathcal{A} \times \mathcal{S} \times \mathcal{G} \longmapsto \mathbb{R}$ is deterministic, and $r(s_t, a_t, s_{t+1} | s_g) := \mathbb{I}[s_{t+1} = s_g]$. That is, there is a positive reward when an agent reaches the

goal ($s_{t+1} = s_g$), and $0$ everywhere else. Since the goal is given to the agent at the beginning of the episode, in goal-conditioned RL the agent knows what this task reward function is (unlike the more general RL problem). The transition dynamics are goal-conditioned as well, with an automatic transition to an absorbing state $\bar{s}$ on reaching the goal $s_g$ and then staying in that state with no rewards thereafter ($P(\bar{s}|s_g, a, s_g) = 1 \ \forall \ a \in \mathcal{A}$ and $P(\bar{s}|\bar{s}, a, s_g) = 1 \ \forall \ a \in \mathcal{A}$). In short, the episode terminates on reaching the goal state.

The agent takes actions in this environment based on a policy $\pi \in \Pi : \mathcal{S} \times \mathcal{G} \longmapsto \Delta(\mathcal{A})$. The return $H_g$ for an episode with goal $s_g$ is the discounted cumulative reward over that episode $H_g = \sum_{t=0}^{\infty} \gamma^t r(s_t, a_t, s_{t+1}|s_g)$, where $s_0 \sim \rho_0$, $a_t \sim \pi(\cdot|s_t, s_g)$, and $s_{t+1} \sim P(\cdot|s_t, a_t, s_g)$. The agent aims to find the policy $\pi^* = \arg\max_{\pi \in \Pi} \mathbb{E}_{g \in \mathcal{G}} \mathbb{E}_{s_0 \sim \rho_0} \mathbb{E}_\pi[H_g]$ that maximizes the expected returns in this goal-conditioned MDP. For a policy $\pi$, the agent's goal-conditioned state distribution $\rho_\pi(s|s_g) = \mathbb{E}_{s_0 \sim \rho_0}[(1-\gamma) \sum_{t=0}^{\infty} \gamma^t P(s_t = s|\pi, s_g)]$. Overloading the terminology a bit, we also define the goal-conditioned target distribution $\rho_g(s|s_g) = \delta(s_g)$, a Dirac measure at the goal state $s_g$.

While learning using traditional RL paradigms is possible in goal-conditioned RL, there has also been previous work (Section 2.2) on leveraging the structure of the problem across goals. Hindsight Experience Replay (HER) [1] attempts to speed up learning in this sparse reward setting by taking episodes of agent interactions, where they might not have reached the goal specified for that episode, and relabeling the transitions with the goals that *were* achieved during the episode. Off-policy learning algorithms are then used to learn from this relabeled experience.

## 3.2 Optimal Transport and Wasserstein-1 Distance

The theory of optimal transport [78, 14] considers the question of how much work must be done to transport one distribution to another optimally, where this notion of work is defined by the use of a ground metric $d$. More concretely, consider a metric space $(\mathcal{M}, d)$ where $\mathcal{M}$ is a set and $d$ is a metric on $\mathcal{M}$ (Definitions in Appendix A). For two distributions $\mu$ and $\nu$ with finite moments on the set $\mathcal{M}$, the Wasserstein-$p$ distance is denoted by:

$$W_p(\mu, \nu) := \inf_{\zeta \in Z(\mu,\nu)} \mathbb{E}_{(X,Y) \sim \zeta}[d(X, Y)^p]^{1/p} \tag{1}$$

where $Z$ is the space of all possible couplings, i.e. joint distributions $\zeta \in \Delta(\mathcal{M} \times \mathcal{M})$ whose marginals are $\mu$ and $\nu$ respectively. Finding this optimal coupling tells us what is the least amount of work, as measured by $d$, that needs to be done to convert $\mu$ to $\nu$. This Wasserstein-$p$ distance can then be used as a cost function (negative reward) by an RL agent to match a given target distribution [79, 19, 27].

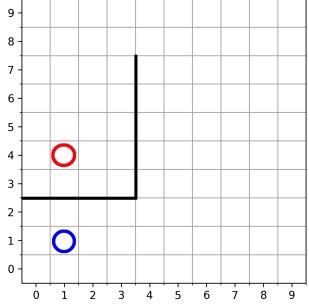

Figure 1: Grid world example

Finding the ideal coupling above is generally considered intractable. However, if what we need is an accurate estimate of the Wasserstein distance and not the optimal transport plan we can turn our attention to the dual form of the Wasserstein-1 distance. The Kantorovich-Rubinstein duality [78, 59] for the Wasserstein-1 distance (which we refer to simply as the Wasserstein distance hereafter) on a ground metric $d$ gives us:

$$W_1(\mu, \nu) = \sup_{\text{Lip}(f) \leq 1} \mathbb{E}_{y \sim \nu}[f(y)] - \mathbb{E}_{x \sim \mu}[f(x)] \tag{2}$$

where the supremum is over all 1-Lipschitz functions $f : \mathcal{M} \longmapsto \mathbb{R}$ in the metric space. Importantly, Jevtić [41] has recently shown that this dual formulation extends to quasimetric spaces as well. More details, such as the definition of the Lipschitz constant of a function and special cases used in WGAN [3, 32, 28] are elaborated in Appendix B. Note that the Lipschitz constant of the potential function $f$ is computed based on the ground metric $d$.

## 4 Time-Step Metric

The choice of the ground metric $d$ is important when computing the Wasserstein distance between two distributions. That is, if we want the Wasserstein distance to give an estimate of the work needed to transport the agent's state visitation distribution to the goal state, the ground metric should incorporate a notion of this work.

Consider the grid world shown in Figure 1, where a wall (bold line) marks an impassable barrier in part of the state space. If the states are specified by their Cartesian coordinates on the grid, the Manhattan distance between the states specified by the blue and red circles is not representative of the optimal cost to go from one to the other. This mismatch would lead to an underestimation of the work involved if the two distributions compared were concentrated at those two circles. Similarly, there will be errors in estimating the Wasserstein distance if the grid world is toroidal (where an agent is transported to the opposite side of the grid if it walks off one side) or if the transitions are asymmetric (windy grid world [74]).

To estimate the work needed to transport measure in an MDP when executing a policy $\pi$, we consider a *quasimetric* – a metric that does not need to be symmetric – dependent on the number of transitions experienced before reaching the goal when executing that policy.

**Definition 1.** *The **time-step metric** $d_T^\pi$ in an MDP with state space $\mathcal{S}$, action space $\mathcal{A}$, transition function $P$, and agent policy $\pi$ is a quasimetric where the distance from state $s \in \mathcal{S}$ to state $s_g \in \mathcal{S}$ is based on the expected number of transitions under policy $\pi$.*

$$d_T^\pi(s, s_g) := \mathbb{E}\ [T(s_g|\pi, s)]$$

*where $T(s_g|\pi, s)$ is the random variable for the first time-step that state $s_g$ is encountered by the agent after starting in state $s$ and following policy $\pi$.*

This quasimetric has the property that the per step cost is uniformly $1$ for all transitions except ones from the goal to the absorbing state (and the absorbing state to itself), which are $0$. Thus, it can be written recursively as:

$$d_T^\pi(s, s_g) = \begin{cases} 0 & \text{if } s = s_g \\ 1 + \mathbb{E}_{a \sim \pi(\cdot|s,s_g)}\, \mathbb{E}_{s' \sim P(\cdot|s,a,s_g)}\, [d_T^\pi(s', s_g)] & \text{otherwise} \end{cases} \quad (3)$$

Recall that in order to estimate the Wasserstein distance using the dual (Equation 2) in a metric space where the ground metric $d$ is this time-step metric, the potential function $f : \mathcal{S} \longmapsto \mathbb{R}$ needs to be 1-Lipschitz with respect to $d_T^\pi$. In Appendix C we prove that $L$-Lipschitz continuity can be ensured by enforcing that the difference in values of $f$ on expected transitions from every state are bounded by $L$, implying

$$\mathrm{Lip}(f) \leq \sup_{s \in \mathcal{S}} \left\{ \mathbb{E}_{a \sim \pi(\cdot|s,s_g)} \mathbb{E}_{s' \sim P(\cdot|s,a,s_g)} [|f(s) - f(s')|] \right\}. \quad (4)$$

Note that finding a proper way to enforce the Lipschitz constraint in adversarial methods remains an open problem [49]. However, for the time-step metric considered here, equation 4 is one elegant way of doing so. By ensuring that the Kantorovich potentials do not drift too far from each other on expected transitions under agent policy $\pi$ in the MDP, the conditions necessary for the potential function to estimate the Wasserstein distance can be maintained [78, 3]. Finally, the minimum distance $d_T^\blacklozenge$ from state $s$ to a given goal state $s_g$ (corresponding to policy $\pi^\blacklozenge$) is defined by the Bellman optimality condition (Equation 16 in Appendix D).

Consider how the time-step distance to the goal and the value function for goal-conditioned RL relate to each other. When the reward is $0$ everywhere except for transitions to the goal state, the value becomes $V^\pi(s|s_g) = \mathbb{E}\left[\gamma^{T(s_g|\pi,s)}\right]$. $d_T^\pi(s_0, s_g)$ and $V(s_0|s_g)$ are related as follows.

**Proposition 1.** *A lower bound on the value of any state under a policy $\pi$ can be expressed in terms of the time-step distance from that state to the goal: $V(s_0|s_g) \geq \gamma^{d_T^\pi(s_0,s_g)}$.*

The proofs for all theoretical results are in Appendix D. The Jensen gap $\Delta_{\text{Jensen}}^\pi(s) := V^\pi(s|s_g) - \gamma^{d_T^\pi(s,s_g)}$ describes the sharpness of the lower bound in the proposition above and it is zero if and only if $\mathrm{Var}(T(s_g|\pi, s)) = 0$ [47]. From this line of reasoning, we deduce the following proposition:

**Proposition 2.** *If the transition dynamics are deterministic, the policy that maximizes expected return is the policy that minimizes the time-step metric ($\pi^* = \pi^\blacklozenge$).*

## 5 Wasserstein-1 Distance for Goal-Conditioned Reinforcement Learning

In this section we consider the problem of goal-conditioned reinforcement learning. In Section 5.1 we analyze the Wasserstein distance computed under the time-step metric $d_T^\pi$. Section 5.2 proposes an

algorithm, Adversarial Intrinsic Motivation (AIM), to learn the potential function for the Kantorovich-Rubinstein duality used to estimate the Wasserstein distance, and giving an intrinsic reward function used to update the agent policy in tandem.

## 5.1 Wasserstein-1 Distance under the Time-Step Metric

From Sections 3.2 and 4 the Wasserstein distance under the time-step metric $d_T^\pi$ of an agent policy $\pi$ with visitation measure $\rho_\pi$ to a particular goal $s_g$ and its distribution $\rho_g$ can be expressed as:

$$W_1^\pi(\rho_\pi, \rho_g) = \sum_{s \in S} \rho_\pi(s|s_g) d_T^\pi(s, s_g) \tag{5}$$

where $W_1^\pi$ refers to the Wasserstein distance with the ground metric $d_T^\pi$.

The following proposition shows that the Wasserstein distance decreases as $d_T^\pi(s, s_g)$ decreases, while also revealing a surprising connection with the Jensen gap.

**Proposition 3.** *For a given policy $\pi$, the Wasserstein distance of the state visitation measure of that policy from the goal state distribution $\rho_g$ under the ground metric $d_T^\pi$ can be written as*

$$W_1^\pi(\rho_\pi, \rho_g) = \mathop{\mathbb{E}}_{s_0 \sim \rho_0} \left[ h(d_T^\pi(s_0, s_g)) + \frac{\gamma}{1-\gamma} (\Delta_{Jensen}^\pi(s_0) - 1) \right] \tag{6}$$

*where $h$ is an increasing function of $d_T^\pi$.*

The first component in the above analytical expression shows that the Wasserstein distance depends on the expected number of steps, decreasing if the expected distance decreases. The second component shows the risk-averse nature of the Wasserstein distance. Concretely, the bounds for the Jensen inequality given by Liao and Berg [47] imply that there are non-negative constants $C_1 = C_1(d_T^\pi, \gamma)$ and $C_2 = C_2(d_T^\pi, \gamma)$ depending only on the expected distance and discount factor such that

$$C_1 \text{Var}(T(s_g|\pi, s)) \leq \Delta_{Jensen}^\pi(s) \leq C_2 \text{Var}(T(s_g|\pi, s)).$$

From the above, we can deduce that a policy with lower variance will have lower Wasserstein distance when compared to a policy with same expected distance from the start but higher variance. The relation between the optimal policy in goal-conditioned RL and the Wasserstein distance can be made concrete if we consider deterministic dynamics.

**Theorem 1.** *If the transition dynamics are deterministic, the policy that minimizes the Wasserstein distance over the time-step metrics in a goal-conditioned MDP (see equation 5) is the optimal policy.*

## 5.2 Adversarial Intrinsic Motivation to minimize Wasserstein-1 Distance

The above section makes it clear that minimizing the Wasserstein distance to the goal will lead to a policy that reaches the goal in as few steps as possible in expectation. If the dynamics of the MDP are deterministic, this policy will also be optimal. Note that the dual form (Equation 2) can be used to estimate the distance, *even if the ground metric $d_T^\pi$ is not known*. The smoothness requirement on the potential function $f$ can be ensured with the constraint in Equation 4 on all states and subsequent transitions expected under the agent policy.

Now consider the full problem. The reinforcement learning algorithm aims to learn a goal-conditioned policy with parameters $\theta \in \Theta$ whose state visitation distribution $\rho_\theta$ minimizes the Wasserstein distance to a goal-conditioned target distribution $\rho_g$ for a given goal $s_g \sim \sigma$. AIM leverages the presence of the set of goals that the agent should be capable of reaching with a goal-conditioned potential function $f_\phi : S \times G \longmapsto \mathbb{R}$ with parameters $\phi \in \Phi$. These objectives of the potential function and the agent can be expressed together using the following adversarial objective:

$$\min_{\theta \in \Theta} \max_{\phi \in \Phi} \mathop{\mathbb{E}}_{s_g \sim \sigma} \left[ f_\phi(s_g, s_g) - \mathop{\mathbb{E}}_{s \sim \rho_\theta} [f_\phi(s, s_g)] \right] \tag{7}$$

where the potential function $f_\phi$ is 1-Lipschitz over the state space. Combining the objectives in Equations 7 and 4, the loss for the potential function $f_\phi$ then becomes:

$$
\begin{aligned}
L_f := \mathop{\mathbb{E}}_{s_g \sim \sigma} & \left[ -f_\phi(s_g, s_g) + \mathop{\mathbb{E}}_{s \sim \rho_\theta} [f_\phi(s, s_g)] \right] + \\
& \lambda \mathop{\mathbb{E}}_{(s,a,s',s_g) \sim \mathcal{D}} \left[ (\max(|f_\phi(s, s_g) - f_\phi(s', s_g)| - 1, 0))^2 \right]
\end{aligned}
\tag{8}
$$

Where the distribution $\mathcal{D}$ should ideally contain all states in $\mathcal{S}$, expected goals in $\mathcal{G}$, and the transitions according to the agent policy $\pi_\theta$ and transition function $P$. Such a distribution is difficult to obtain directly. AIM approximates it with a small replay buffer of transitions from recent episodes experienced by the agent, and relabels these episodes with achieved goals (similar to HER [1]). Such an approximation does not respect the discounted measure of states later on in an episode, but is consistent with how other approaches in deep reinforcement learning tend to approximate the state visitation distribution, especially for policy gradient approaches [55]. While it does not include all states and all goals, we see empirically that the above approximation works well.

Now we turn to the reward function that should be presented to the agent to minimize the Wasserstein distance. The Wasserstein discriminator is a potential function [52] (its value depends on the state). It can thus be used to create a shaped reward $\hat{r}(s, a, s', s_g) = r(s, a, s'|s_g) + \gamma f_\phi(s', s_g) - f_\phi(s, s_g)$ without risk of changing the optimal policy. Alternatively, we can explicitly minimize samples of the Wasserstein distance: $\hat{r}(s, a, s', s_g) = f_\phi(s', s_g) - f_\phi(s_g, s_g)$. Finally, instead of the second term $f_\phi(s_g, s_g)$, we can just use a constant bias term. In practice, all these choices work well, and the experiments use the latter (with $b = \max_{s \in \mathcal{S}} f_\phi(s, s_g)$) to reduce variance in $\hat{r}$.

$$\hat{r}(s, a, s', s_g) = f_\phi(s', s_g) - b \tag{9}$$

The basic procedure to learn and use adversarial intrinsic motivation (AIM) is laid out in Algorithm 1, and also includes how to use this algorithm in conjunction with HER. If not using HER, Line 23 where hindsight goals are added to the replay buffer can be skipped.

## 6 Experiments

Our experiments evaluate the extent to which the reward learned through AIM is useful as a proxy for the environment reward signal, or in tandem with the environment reward signal. In particular, we ask the following questions:

- Does AIM speed up learning of a policy to get to a single goal compared to learning with a sparse reward?
- Does the learned reward function qualitatively guide the agent to the goal?
- Does AIM work well with stochastic transition dynamics or sharp changes in the state features?
- Does AIM generalize to a large set of goals and continuous state and action spaces?

Our experiments suggest that the answer to all 4 questions is "yes", with the first three questions tested in the grid world presented in Figure 1 where the goal is within a room, and the agent has to go around the room from its start state to reach the goal. Goal-conditioned tasks in the established Fetch robot domain show that AIM also accelerates learning across multiple goals in continuous state and action spaces.

This section compares an agent learning with a reward learned through AIM with other intrinsic motivation signals that induce general exploration or shaped rewards that try to guide the agent to the goal. The experiments show that AIM guides the agent's exploration more efficiently and effectively than a general exploration bonus, and adapts to the dynamics of the environment better than other techniques we compare to. As an overview, the baselines we compare to are:

- **RND**: with random network distillation (RND) [16] used to provide a general exploration bonus.
- **MC**: with the distance between states learned through regression of Monte Carlo rollouts of the agent policy, similar to Hartikainen et al. [35].
- **SMiRL**: SMiRL [13] is used to provide a bonus intrinsic motivation reward that minimizes the overall surprise in an episode.
- **DiscoRL** The DiscoRL [51] approach presents a reward to maximize the likelihood of a target distribution (normal distribution at the goal). In practice this approach is equivalent to a negative L2 distance to the goal, which we compare to in the grid world domain.
- **GAIL**: additional GAIL [39] rewards using trajectories relabeled with achieved goals considered as having come from the expert in hindsight. This baseline is compared to in the Fetch robot domain, since that is the domain where we utilize hindsight relabeling.

**Grid World**  In this task, the goal is inside a room and the agent's starting position is such that it needs to navigate around the room to find the doorway and be able to reach the goal. The agent can

move in the 4 cardinal directions unless blocked by a wall or the edge of the grid. The agent policy is learned using soft Q-learning [33] with no hindsight goals used for this experiment.

The agent's state visitation distribution after just 100 Q-function updates when using AIM-learned rewards is shown in Figure 2a and the learned rewards for each state are plotted in Figure 2b. The state visitation when learning with the true task reward shows that the agent is unable to learn a policy to the goal (Figure 2c). These figures show that AIM enables the agent to reach the goal and learn the required policy quickly, while learning with the sparse task reward fails to do so.

In Appendix F we also compare to the baselines described above and show that AIM learns a reward that is more efficient at directing the agent's exploration and more flexible to variations of the environment dynamics, such as stochastic dynamics or transitions that cause a sharp change in the state features. None of the baselines compared to were able to direct the agent to the goal in this grid world even after given up to 5× more interactions with the environment to train. AIM's use of the time-step metric also enabled it to adapt to variations of the environment dynamics better than the gradient penalty based regularization used in Wasserstein GANs [32] and adversarial imitation learning [28] approaches.

**Fetch Robot**   The generalization capability of AIM across multiple goals in goal-conditioned RL tasks with continuous states and actions is tested in the MuJoCo simulator [75], on the Fetch robot tasks from OpenAI gym [15] which have been used to evaluate learning of goal-conditioned policies previously [1, 80]. Descriptions of these tasks and their goal space is in Appendix H. We soften the Dirac target distribution for continuous states to instead be a Gaussian with variance of 0.01 of the range of each feature.

The goals in this setting are not the full state, but rather the dimensions of factored states relevant to the given goal. The task wrapper additionally returns the features of the agent's state in this reduced goal space, and so AIM can use it to learn our reward function, rather than the full state space. It is unclear how this smaller goal space might affect AIM. While the smaller goal space might make learning easier for potential function $f_\phi$, the partially observable nature of the goals might lead to a less informative reward.

We combine AIM with HER (refer subsection 3.1) and refer to it as [AIM + HER]. We compare this agent to the baselines we referred to above, as well as the sparse environment reward (R + HER) and the dense reward derived from the negative Euclidean ($L2$) distance to the goal ($-L2$ + HER). The $L2$ distance is proportional to the number of steps it should take the agent to reach the goal in this environment, and so the reward based on it can act as an oracle reward that we can use to test how efficiently AIM can learn a reward function that helps the agent learn its policy. We used the HER implementation using Twin Delayed DDPG (TD3) [26] as the underlying RL algorithm from the stable baselines repository [38]. We did an extensive sweep of the hyperparameters for the baseline HER + R (laid out in Appendix H), with a coarser search on relevant hyperparameters for AIM.

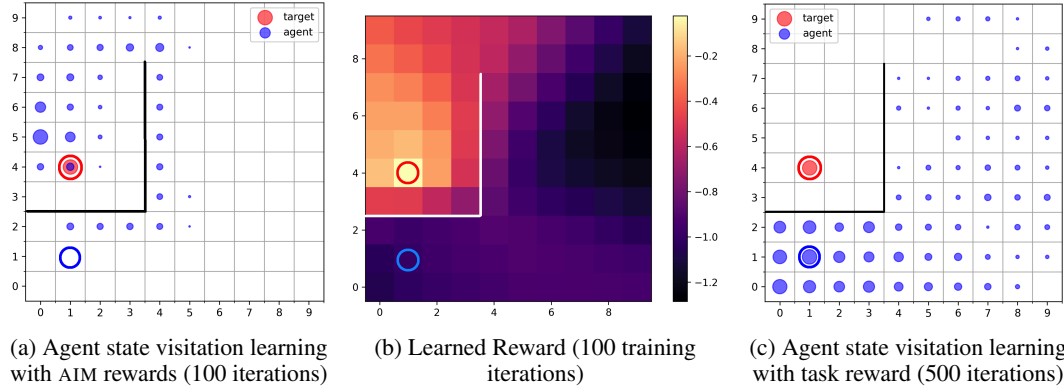

(a) Agent state visitation learning with AIM rewards (100 iterations)

(b) Learned Reward (100 training iterations)

(c) Agent state visitation learning with task reward (500 iterations)

Figure 2: Grid world experiments. Agent's undiscounted state visitation (2a, 2c): Blue circle indicates agent's start state. Red circle is the goal. Blue bubbles indicate relative time agent's policy causes it to spend in respective states. Learned reward function (2b): AIM reward at each state of the grid world. Bold black (or white) lines indicate walls the agent cannot transition through.

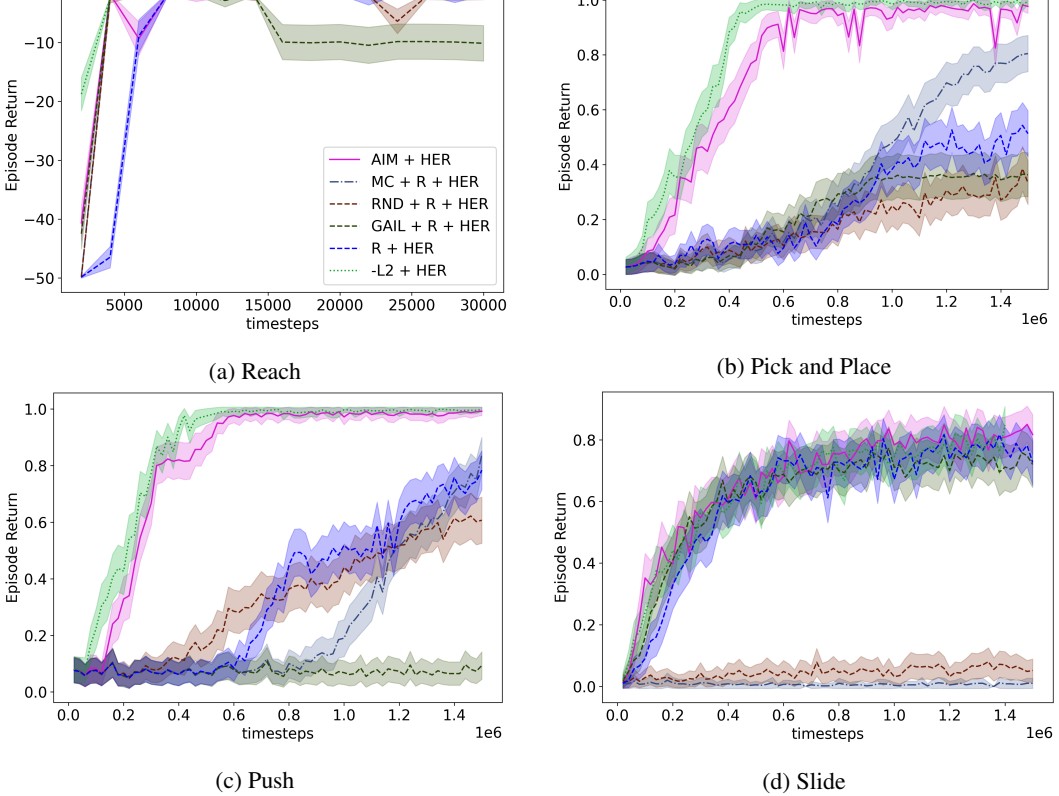

(a) Reach

(b) Pick and Place

(c) Push

(d) Slide

Figure 3: Evaluating AIM with HER on some goal-conditioned RL tasks in the Fetch domain. AIM learns the reward function in tandem with the policy updates. The "$-L2$" reward is the true negative distance to goal, acting as a oracle reward in this domain. The other baselines are detailed above.

Figure 3 shows that using the AIM-learned reward speeds up learning in three of the four Fetch domains, even without the environment reward. This improvement is very close to what we would see if we used the dense reward (based on the actual distance function in this domain). An additional comparison with an agent learning with both the AIM-learned reward as well as the task reward (AIM + R + HER) can be seen in Figure 7 in the Appendix, showing that using both signals accelerates learning even more. These results also highlight that AIM continues to work in continuous state and action spaces, even though our analysis focuses on discrete states and actions. Results are averaged across the 6 different seeds, with shaded regions showing standard error across runs. Statistical analysis using a mixed effects ANOVA and a Tukey test at a significance level of $95\%$ (more detail in Appendix G) show that in three of the four environments AIM and AIM+ R have similar odds of reaching the goal as the dense shaped reward, and in all four environments AIM and AIM+ R have higher odds of reaching the goal compared to the sparse reward.

The other baselines compare well to AIM in the Fetch Reach domain (Figure 3a), but do not do as well on the other problems. In fact, none of the other baselines outperform the vanilla baseline [R + HER] in all the domains. The RND rewards help the agent to start learning faster in the Push domain (Figure 3c), but lead to worse performance in Pick and Place (Figure 3b). On the other hand, learning the distance function through MC regression helps in the Pick and Place domain, but slows down learning when dealing with Push. Most notably, both these approaches cause learning to fail in the Slide domain (Figure 3d), where the credit assignment problem is especially difficult. GAIL works as well as AIM and the vanilla baseline in Slide, but underperforms in the other domains. We hypothesize that the additional rewards in these baselines conflict with the task reward. Additionally, none of the three new baselines work well if we do not provide the task reward in addition to the specific bonus for that algorithm.

We did not find any configuration in the Fetch Reach domain where [SMiRL + R + HER] was able to accomplish the task in the given training budget. Since SMiRL did not work on the grid world or Fetch Reach, we did not try it out on any of the other domains.

FAIRL [28] (which has been shown to learn policies that cover hand-specified state distributions) was also applied on these 4 domains but it failed to learn at all. Interestingly, scaling the reward such that it is always negative led to similar performance to (but not better than) AIM. We hypothesize that FAIRL, as defined and presented, fails in these domains because the environments are episodic, and the episode ends earlier if the goal is reached. Since the FAIRL reward is positive closer to the target distribution, the agent can get close to the target, but refrain from reaching it (and ending the episode) to collect additional positive reward.

The domain where AIM does not seem to have a large advantage (Slide) is one where the agent strikes an object initially and that object has to come to rest near the goal. In fact, AIM-learned rewards, the vanilla environment reward R, and the oracle $-L2$ rewards all lead to similar learning behavior, indicating that this particular task does not benefit much from shaped rewards. The reason for this invariance might be that credit assignment has to propagate back to the single point when the agent strikes the object regardless of how dense the subsequent reward is.

## 7    Discussion and Future Work

Approaches for estimating the Wasserstein distance to a target distribution by considering the dual of the Kantorovich relaxation have been previously proposed [3, 32, 79], but assume that the ground metric is the $L2$ distance. We improve upon them by choosing a metric space more suited to the MDP and notions of optimality in the MDP. This choice allows us to leverage the structure introduced by the dynamics of the MDP to regularize the Kantorovich potential using a novel objective.

Previous work [12] has pointed out that the gradients from sample estimates of the Wasserstein distance might be biased. This issue is mitigated in our implementation through multiple updates of the discriminator, which they found to be empirically useful in reducing the bias. Additionally, recent work has pointed out that the discriminator in WGAN might be bad at estimating the Wasserstein distance [73]. While our experiments indicate that the potential function in AIM is learned appropriately, future work could look more deeply to verify possible inefficiencies in this estimation.

The process of learning the Wasserstein distance through samples of the environment while simultaneously estimating the cost of the full path is reminiscent of the $A^*$ algorithm [34], where the optimistic heuristic encourages the agent to explore in a directed manner, and adjusts its estimates based on these explorations.

The discriminator objective (Equation 8) also bears some resemblance to a linear program formulation of the RL problem [60]. The difference is that this formulation minimizes the value function on states visited by the agent, while AIM additionally maximizes the potential at the goal state. This crucial difference has two main consequences. First, the potential function during learning is not equivalent to the value of the agent's policy (verified by using this potential as a critic). Second, increasing the potential of the goal state in AIM directs the agent exploration in a particular direction (namely, the direction of sharpest increase in potential).

In the goal-conditioned RL setting, AIM seems to be an effective intrinsic reward that balances exploration and exploitation for the task at hand. The next step is to consider whether the Wasserstein distance can be estimated similarly for more general tasks, and whether minimizing this distance in those tasks leads to the optimal policy. A different potential avenue for future work is the problem of more general exploration [37, 46] by specifying a uniform distribution as the target, or using this directed exploration as an intermediate step for efficient exploration [42].

Finally, reward design is an important aspect of practical reinforcement learning. Not only do properly shaped reward speed up learning [52], but reward design can also subtly influence the kinds of behaviors deemed acceptable for the RL agent [45] and could be a potential safety issue keeping reinforcement learning from being deployed on real world problems. Learning-based approaches that can assist in specifying reward functions safely given alternative approaches for communicating the task could be of value in such a process of reward design, and an avenue for future research.

## Acknowledgements and Funding Information

We thank Caroline Wang, Garrett Warnell, and Elad Liebman for discussion and feedback on this work. We also thank the reviewers for their thoughtful comments and suggestions that have helped to improve this paper.

This work has taken place in part in the Learning Agents Research Group (LARG) at the Artificial Intelligence Laboratory, and in part in the Personal Autonomous Robotics Lab (PeARL) at The University of Texas at Austin. LARG research is supported in part by the National Science Foundation (CPS-1739964, IIS-1724157, FAIN-2019844), the Office of Naval Research (N00014-18-2243), Army Research Office (W911NF-19-2-0333), DARPA, Lockheed Martin, General Motors, Bosch, and Good Systems, a research grand challenge at the University of Texas at Austin. PeARL research is supported in part by the NSF (IIS-1724157, IIS-1638107, IIS-1749204, IIS-1925082), ONR (N00014-18-2243), AFOSR (FA9550-20-1-0077), and ARO (78372-CS). This research was also sponsored by the Army Research Office under Cooperative Agreement Number W911NF-19-2-0333. The views and conclusions contained in this document are those of the authors and should not be interpreted as representing the official policies, either expressed or implied, of the Army Research Office or the U.S. Government. The U.S. Government is authorized to reproduce and distribute reprints for Government purposes notwithstanding any copyright notation herein. Peter Stone serves as the Executive Director of Sony AI America and receives financial compensation for this work. The terms of this arrangement have been reviewed and approved by the University of Texas at Austin in accordance with its policy on objectivity in research.

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
