# A  Metrics and Quasimetrics

A metric space $(\mathcal{M}, d)$ is composed of a set $\mathcal{M}$ and a metric $d : \mathcal{M} \times \mathcal{M} \longmapsto \mathbb{R}^+ \cup \{\infty\}$ that compares two points in that set. Here $\mathbb{R}^+$ is the set of non-negative real numbers.

**Definition 2.** *A metric $d : \mathcal{M} \times \mathcal{M} \longmapsto \mathbb{R}^+ \cup \{\infty\}$ compares two points in set $\mathcal{M}$ and satisfies the following axioms $\forall m_1, m_2, m_3 \in \mathcal{M}$:*

- $d(m_1, m_2) = 0 \iff m_1 = m_2$ *(identity of indiscernibles)*
- $d(m_1, m_2) = d(m_2, m_1)$ *(symmetry)*
- $d(m_1, m_2) \leq d(m_1, m_3) + d(m_3, m_2)$ *(triangle inequality)*

A variation on metrics that is important to this paper is *quasimetrics*.

**Definition 3.** *A quasimetric [66] is a function that satisfies all the properties of a metric, with the exception of symmetry $d(m_1, m_2) \neq d(m_2, m_1)$.*

As an example, consider an MDP where the actions and transition dynamics allow an agent to navigate from any state to any other state. Let $T(s_2|\pi, s_1)$ be the random variable for the first time-step that state $s_2$ is encountered by the agent after starting in state $s_1$ and following policy $\pi$. The time-step metric $d_T^\pi$ for this MDP can then be defined as

$$d_T^\pi(s_1, s_2) := \mathbb{E}\ [T(s_2|\pi, s_1)]$$

$d_T^\pi$ is a quasimetric, since the action space and transition function need not be symmetric, meaning the expected minimum time needed to go from $s_1$ to $s_2$ need not be the same as the expected minimum time needed to from $s_2$ to $s_1$. The diameter of an MDP [39, 43] is generally calculated by taking the maximum time-step distance between over all pairs of states in the MDP either under a random policy or a policy that travels from any state to any other state in as few steps as possible.

# B  Optimal Transport and Wasserstein-1 Distance

The theory of optimal transport [74, 14] considers the question of how much work must be done to transport one distribution to another optimally. More concretely, suppose we have a metric space $(\mathcal{M}, d)$ where $\mathcal{M}$ is a set and $d$ is a metric on $\mathcal{M}$. See the definitions of metrics and quasimetrics in Appendix A. For two distributions $\mu$ and $\nu$ with finite moments on the set $\mathcal{M}$, the Wasserstein-$p$ distance is denoted by:

$$W_p(\mu, \nu) := \inf_{\zeta \in Z(\mu, \nu)} \mathbb{E}_{(X,Y) \sim \zeta}\left[d(X, Y)^p\right]^{1/p} \tag{10}$$

where $Z$ is the space of all possible couplings between $\mu$ and $\nu$. Put another way, $Z$ is the space of all possible distributions $\zeta \in \Delta(\mathcal{M} \times \mathcal{M})$ whose marginals are $\mu$ and $\nu$ respectively. Finding this optimal coupling tells us what is the least amount of work, as measured by $d$, that needs to be done to convert $\mu$ to $\nu$. This Wasserstein-$p$ distance can then be used as a cost function (negative reward) by an RL agent to match a given target distribution [75, 19].

Finding the ideal coupling (meaning finding the optimal transport plan from one distribution to the other) which gives us an accurate distance is generally considered intractable. However, if what we need is an accurate estimate of the Wasserstein distance and not the optimal transport plan (as is the case when we mean to use this distance as part of our intrinsic reward) we can turn our attention to the dual form of this distance. The Kantorovich-Rubinstein duality [74] for the Wasserstein-1 distance on a ground metric $d$ is of particular interest and gives us the following equality:

$$W_1(\mu, \nu) = \sup_{\text{Lip}(f) \leq 1} \mathbb{E}_{y \sim \nu}\left[f(y)\right] - \mathbb{E}_{x \sim \mu}\left[f(x)\right] \tag{11}$$

where the supremum is over all 1-Lipschitz functions $f : \mathcal{M} \longmapsto \mathbb{R}$ in the metric space, and the Lipschitz constant of a function $f$ is defined as:

$$\text{Lip}(f) := \sup \left\{ \frac{|f(y) - f(x)|}{d(x,y)} \forall (x,y) \in \mathcal{M}^2, x \neq y \right\} \tag{12}$$

That is, the Lipschitz condition of this function $f$ (called the Kantorovich potential function) is measured according to the metric $d$. Recently, Jevtić [40] has shown that this dual formulation where the constraint on the potential function is a smoothness constraint extends to quasimetric spaces as well. If defined over a quasimetric space, the Wasserstein distance also has properties of a quasimetric (specifically, the distances are not necessarily symmetric).

If the given metric space is a Euclidean space ($d(x,y) = \|y - x\|_2$), the Lipschitz bound in Equation 2 can be computed locally as a uniform bound on the gradient of $f$.

$$W_1(\mu, \nu) = \sup_{\|\nabla f\| \leq 1} \mathbb{E}_{y \sim \nu}\left[f(y)\right] - \mathbb{E}_{x \sim \mu}\left[f(x)\right] \tag{13}$$

meaning that $f$ is the solution to an optimization objective with the restriction that $\|\nabla f(x)\| \leq 1$ for all $x \in \mathcal{M}$. This strong bound on the dual in Euclidean space is the one that has been used most in recent implementations of the Wasserstein generative adversarial network [3, 31] to regularize the learning of the discriminator function. Such regularization has been found to be effective for stability in other adversarial learning approaches such as adversarial imitation learning [27].

Practically, the Kantorovich potential function $f$ can be approximated using samples from the two distributions $\mu$ and $\nu$, regularization of the potential function to ensure smoothness, and an expressive function approximator such as a neural network. A more in depth treatment of the Kantorovich relaxation and the Kantorovich-Rubinstein duality, as well as their application in metric and Euclidean spaces using the Wasserstein-1 distance we lay out above, is provided by Peyré and Cuturi [57].

Now consider the problem of goal-conditioned reinforcement learning. Here the target distribution $\nu$ is the goal-conditioned target distribution $\rho_g$ which is a Dirac at the given goal state. Similarly, the distribution to be transported $\mu$ is the agent's goal-conditioned state distribution $\rho_\pi$.

The Wasserstein-1 distance of an agent executing policy $\pi$ to the goal $s_g$ can be expressed in a fairly straightforward manner as:

$$W_1(\rho_\pi, \rho_g) = \sum_{s \in \mathcal{S}} \rho_\pi(s|s_g) d(s, s_g) \tag{14}$$

The above is a simplification of Equation 1, where $p = 1$ and the joint distribution is easy to specify since the target distribution $\rho_g$ is a Dirac distribution.

## C  Lipschitz constant of Potential function

For a given goal $s_g$ and all states $s_0 \in \mathcal{S}$, recall that function $f$ is $L$-Lipschitz if it follows the Lipschitz condition as follows.

$$|f(s_g) - f(s_0)| \leq L d_T^\pi(s_0, s_g) \; \forall s_0 \in \mathcal{S} \tag{15}$$

**Proposition 4.** *If transitions from the agent policy $\pi$ are guaranteed to arrive at the goal in finite time and $f$ is $L$-bounded in expected transitions, i.e.,*

$$\sup_{s \in S} \mathbb{E}_{s' \sim \pi, P} \left[|f(s') - f(s)|\right] \leq L,$$

*then $f$ is $L$-Lipschitz.*

*Proof.* Since $f(s_g) - f(s_0)$ is a scalar quantity, we may write $f(s_g) - f(s_0) = \mathbb{E}_{\pi, P}[f(s_g) - f(s_0)]$. Using this fact and that $P(T(s_0) < \infty) = 1$ where $T(s_0) = T^\pi(s_g|\pi, s_0)$ for notation simplicity,

the LHS of the expression above becomes a telescopic sum

$$|f(s_g) - f(s_0)| = \mathop{\mathbb{E}}_{\pi,P} [f(s_g) - f(s_0)]$$

$$= \mathop{\mathbb{E}}_{\pi,P} \left[ \left| \sum_{t=0}^{T(s_0)-1} (f(s_{t+1}) - f(s_t)) \right| \right] .$$

$$\leq \mathop{\mathbb{E}}_{\pi,P} \left[ \sum_{t=0}^{T(s_0)-1} |f(s_{t+1}) - f(s_t)| \right] .$$

Now let us assume that for all transitions $(s, a, s')$, $\mathbb{E}[|f(s') - f(s)|] \leq L$. Then

$$\mathop{\mathbb{E}}_{\pi,P} \left[ \sum_{t=0}^{T(s_0)-1} |f(s_{t+1}) - f(s_t)| \right] = \mathop{\mathbb{E}}_{T(s_0)} \left[ \mathop{\mathbb{E}}_{\pi,P} \left[ \sum_{t=0}^{T(s_0)-1} |f(s_{t+1}) - f(s_t)| \Big| T(s_0) \right] \right]$$

$$\leq \mathop{E}_{T(s_0)} \left[ \sum_{t=0}^{T(s_0)-1} L \right]$$

$$= L \mathop{\mathbb{E}}_{T(s_0)} [T(s_0)]$$

$$= L d_T^\pi(s_0, s_g),$$

showing that $|f(s_g) - f(s_0)| \leq L d_T^\pi(s_0, s_g)$ as desired. $\square$

# D Proofs of Claims

The Bellman optimality condition gives us the following optimal distance to goal:

$$d_T^\blacklozenge(s, s_g) = \begin{cases} 0 & \text{if } s = s_g \\ 1 + \min_{a \in \mathcal{A}} \sum_{s' \in \mathcal{S}} P(s'|s, a, s_g) d_T^\blacklozenge(s', s_g) & \text{otherwise} \end{cases} \tag{16}$$

**Proposition 1.** *A lower bound on the value of any state under a policy $\pi$ can be expressed in terms of the time-step distance from that state to the goal: $V(s_0|s_g) \geq \gamma^{d_T^\pi(s_0, s_g)}$.*

*Proof.*

$$V^\pi(s|s_g) = \mathbb{E}\left[ \gamma^{T(s_g|\pi, s)} \right] \geq \gamma^{d_T^\pi(s, s_g)} \quad \forall \, s \in \mathcal{S}$$

where the inequality follows as a consequence of Jensen's inequality and the convex nature of the value function. $\square$

**Proposition 2.** *If the transition dynamics are deterministic, the policy that maximizes expected return is the policy that minimizes the time-step metric ($\pi^* = \pi^\blacklozenge$).*

*Proof.* Consider the value of a state $s$ given goal $s_g$. If the transitions are deterministic and the agent policy $\pi$ is deterministic (as is the case for the optimal policy), then the time to reach the goal satisfies $\text{Var}(T(s_g|\pi, s)) = 0$, implying that $\Delta_{\text{Jensen}}$ vanishes and therefore

$$V^\pi(s|s_g) = \gamma^{d_T^\pi(s, s_g)}.$$

Since $\gamma \in [0, 1)$, $V^\pi$ is monotonically decreasing with $d_T^\pi$

$$\arg\max_\pi V^\pi(s|s_g) = \arg\min_\pi d_T^\pi(s, s_g) \, \forall \, s \in \mathcal{S}$$

That is, in the deterministic transition dynamics scenario, $\pi^* = \pi^\blacklozenge$. $\square$

**Proposition 3.** *For a given policy $\pi$, the Wasserstein distance of the state visitation measure of that policy from the goal state distribution $\rho_g$ under the ground metric $d_T^\pi$ can be written as*

$$W_1^\pi(\rho_\pi, \rho_g) = \mathop{\mathbb{E}}_{s_0 \sim \rho_0} \left[ h(d_T^\pi(s_0, s_g)) + \frac{\gamma}{1 - \gamma}(\Delta_{Jensen}^\pi(s_0) - 1) \right] \tag{6}$$

*where $h$ is an increasing function of $d_T^\pi$.*

*Proof.* The first step of the proof is to obtain an analytical expression for the the expected distance to the goal after $t$ steps as a function of the expected distance at $t = 0$. To reduce the notation burden, denote $T(s_0) = T(s_g|\pi, s_0)$ and let $s_t(s_0)$ be the state after $t$ steps conditional on some starting state $s_0$ where actions are taken according to $\pi$. We have excluded $s_g$ and $\pi$ from the notation since they are fixed for the purpose of this proposition. Using the law of total expectation we have that for every initial $s_0$

$$\mathbb{E}_{s_t}[d(s_t(s_0), s_g)] = \mathbb{E}_{T(s_0)}[\mathbb{E}_{s_t}[d(s_t(s_0), s_g) \mid T(s_0)]] = \mathbb{E}_{T(s_0)}[\max(T(s_0) - t, 0)],$$

Now, by expanding the definition of $\rho_\pi(s \mid s_g)$ in equation 5, exchanging the order of summation, and using the previous equation we may write

$$W_1^\pi(\rho_\pi, \rho_g) = \sum_{s \in \mathcal{S}} \sum_{t=0}^{\infty} (1 - \gamma)\gamma^t \mathbb{E}_{s_0}[P(s_t = s \mid \pi, s_g)]d_T^\pi(s, s_g)$$

$$= \mathbb{E}_{s_0} \left[ (1 - \gamma) \sum_{t=0}^{\infty} \gamma^t \mathbb{E}_{s_t}[d(s_t(s_0), s_g) \mid s_0] \right]$$

$$= \mathbb{E}_{s_0} \left[ \mathbb{E}_{T(s_0)} \left[ (1 - \gamma) \sum_{t=0}^{\infty} \gamma^t \max(T(s_0) - t, 0) \Big| s_0 \right] \right]$$

Standard but tedious algebraic manipulations given in Lemma 1 in the Appendix show that

$$\sum_{t=0}^{\infty} (1 - \gamma)\gamma^t \max(T(s_0) - t, 0) = T(s_0) - \frac{\gamma}{1 - \gamma}(1 - \gamma^{T(s_0)}).$$

Combining the two identities above we arrive at

$$W_1^\pi(\rho_\pi, \rho_g) = \mathbb{E}_{s_0} \left[ \mathbb{E}_{T(s_0)} \left[ T(s_0) - \frac{\gamma}{1 - \gamma}(1 - \gamma^{T(s_0)}) \Big| s_0 \right] \right]$$

$$= \mathbb{E}_{s_0} \left[ d(s_0, s_g) - \frac{\gamma}{1 - \gamma}(1 - \mathbb{E}[\gamma^{T(s_0)} \mid s_0]) \right]$$

$$= \mathbb{E}_{s_0} \left[ d(s_0, s_g) + \frac{\gamma}{1 - \gamma}\gamma^{d(s_0, s_g)} - \frac{\gamma}{1 - \gamma}(1 - \mathbb{E}[\gamma^{T(s_0)} \mid s_0] + \gamma^{d(s_0, s_g)}) \right] \tag{17}$$

$$= \mathbb{E}_{s_0} \left[ d(s_0, s_g) + \frac{\gamma}{1 - \gamma}\gamma^{d(s_0, s_g)} + \frac{\gamma}{1 - \gamma}(\Delta_{Jensen}^\pi(s_0) - 1) \right].$$

To finalize the proof, we only need to show that the function $h(\mu) = \mu + \frac{\gamma}{1-\gamma}\gamma^\mu$ is monotonically increasing for every $\gamma \in [0, 1)$. This is a standard calculus exercise that we show in Lemma 2 in Appendix E. $\qquad\square$

**Theorem 1.** *If the transition dynamics are deterministic, the policy that minimizes the Wasserstein distance over the time-step metrics in a goal-conditioned MDP (see equation 5) is the optimal policy.*

*Proof.* Proposition 2 shows that the Jensen gap vanishes for the optimal policy of an MDP with deterministic transitions and that it minimizes the expected distance from start for all initial states. Proposition 3, on the other hand, implies that when the Jensen gap vanishes, the Wasserstein distance is monotonically increasing in the expected distance from the start. Together, the two propositions show that $\pi^*$ minimizes the Wasserstein distance. $\qquad\square$

**Algorithm 1:** AIM + HER

**Input:** Agent policy $\pi_\theta$, discriminator $f_\phi$, environment $env$,
number of Epochs $N$, number of time-steps per epoch $K$,
policy update period $k$, discriminator update period $m$, episode length $T$,
replay buffer (for HER), smaller replay buffer (for discriminator)

1 Initialize discriminator parameters $\phi$;
2 Initialize policy parameters $\theta$;
3 **for** $n = 0, 1, \ldots, N - 1$ **do**
4      $t = 0$;
5      goal_reached = True;
6      **while** $t < K$ **do**
7          **if** *goal_reached or episode_over* **then**
8              Sample goal $s_g \sim \sigma(\mathcal{G})$;
9              Sample start state $s \sim \rho_0(\mathcal{S})$;
10              goal_reached = False;
11              episode_over = False;
12              $t_{start} = K$;
13          **end**
14          Sample action $a \sim \pi_\theta(\cdot|s, s_g)$;
15          $s' = env.step(a)$;
16          **if** $s' = s_g$ **then**
17              goal_reached = True;
18          **end**
         // end episode if goal not reached in $T$ steps
19          **if** $t - t_{start} = T$ **then**
20              episode_over = True;
21          **end**
22          Add $(s, a, s', s_g, goal\_reached)$ to replay buffer and smaller replay buffer;
23          **if** *goal_reached or episode_over* **then**
24              Add hindsight goals to both buffers;
25          **end**
         // Update policy parameters $\theta$ every $k$ steps
26          **if** $t\%k = 0$ **then**
27              Sample tuples $(s, a, s', s_g, goal\_reached)$ from replay buffer;
28              Get intrinsic reward (Equation 9);
29              Update policy parameters $\theta$ using any off-policy learning algorithm;
30          **end**
         // Update discriminator parameters $\phi$ every $m$ steps
31          **if** $t\%m = 0$ **then**
32              Sample tuples $(s, a, s', s_g, goal\_reached)$ from smaller replay buffer;
33              Update discriminator parameters $\phi$ using Equation 8;
34          **end**
35          $t = t + 1$;
36      **end**
37      Evaluate agent policy;
38 **end**

# E    Auxiliary results for Proposition 3

**Lemma 1.** *Let $T$ be a positive integer. Then*

$$\sum_{t=0}^{\infty}(1 - \gamma)\gamma^t \max(T - t, 0) = T - \frac{\gamma}{1 - \gamma}(1 - \gamma^T).$$

*Proof.* Direct computation gives

$$(1-\gamma)\sum_{t=0}^{\infty}\gamma^t \max(T-t,0) = (1-\gamma)\sum_{t=0}^{T-1}\gamma^t(T-t)$$

$$= (1-\gamma)T\sum_{t=0}^{T-1}\gamma^t - (1-\gamma)\sum_{t=0}^{T-1}t\gamma^t$$

We will now simplify the two terms of the last expression. For the first one, have

$$(1-\gamma)T\sum_{t=0}^{T-1}\gamma^t = (1-\gamma)T\frac{1-\gamma^T}{1-\gamma} = T - T\gamma^T.$$

For the second one, the computations are a bit more involved

$$(1-\gamma)\sum_{t=0}^{T-1}t\gamma^t = (1-\gamma)\gamma\sum_{t=1}^{T-1}t\gamma^{t-1}$$

$$= (1-\gamma)\sum_{t=1}^{T-1}\gamma\frac{d}{d\gamma}\gamma^t$$

$$= \gamma(1-\gamma)\frac{d}{d\gamma}\sum_{t=0}^{T-1}\gamma^t$$

$$= \gamma(1-\gamma)\frac{d}{d\gamma}\frac{1-\gamma^T}{1-\gamma}$$

$$= \frac{\gamma}{(1-\gamma)}\left(-T\gamma^{T-1}(1-\gamma)+(1-\gamma^T)\right) = -T\gamma^T + \frac{\gamma}{(1-\gamma)}(1-\gamma^T).$$

When combining the two simplified expressions the terms with $T\gamma^T$ will cancel out, yielding the desired expression. $\square$

**Lemma 2.** *The function $h_\gamma(\mu) = \mu + \frac{\gamma}{1-\gamma}\gamma^\mu$ is monotonically increasing for every $\gamma \in [0,1)$.*

*Proof.* We must show that $\frac{d}{d\mu}h_\gamma(\mu) > 0$ for every $\gamma \in [0,1)$ and every $\mu > 0$. Computing the derivative directly we obtain

$$\frac{d}{d\mu}h_\gamma(\mu) = 1 + \frac{\log(\gamma)\gamma^{\mu+1}}{1-\gamma}.$$

Thus, it will suffice to show that the second term above is greater than -1. For this purpose, first note that $\log(\gamma)\gamma^{\mu+1} > \log(\gamma)$ since $\gamma < 1$. Now, we use the fact that $\log(\gamma) < 1 - \gamma$ for $\gamma < 1$. This can be verified noting that $1 - \gamma$ is the tangent line to the concave curve $\log(\gamma)$ and the curves meet at $\gamma = 1$. And therefore $\log(\gamma)/(1-\gamma) > -1$. Putting these observation together,

$$\frac{d}{d\mu}h_\gamma(\mu) = 1 + \frac{\log(\gamma)\gamma^{\mu+1}}{1-\gamma} > 1 + \frac{\log(\gamma)}{1-\gamma} > 1 - 1 = 0,$$

concluding the proof. $\square$

## F Grid World Experiments

**Basic experiment** The environment is a $10 \times 10$ grid with $4$ discrete actions that take the agent in the $4$ cardinal directions unless blocked by a wall or the edge of the grid. The agent policy is learned using soft Q-learning [32], with an entropy coefficient of $0.1$ and a discount factor of $\gamma = 0.99$. We do not use hindsight goals for this experiment, and use a single buffer with size $5000$ for both the policy as well as the discriminator training. The results are discussed in the main text. The compute used to conduct these experiments was a personal laptop with an Intel i7 Processor and 16 GB of RAM.

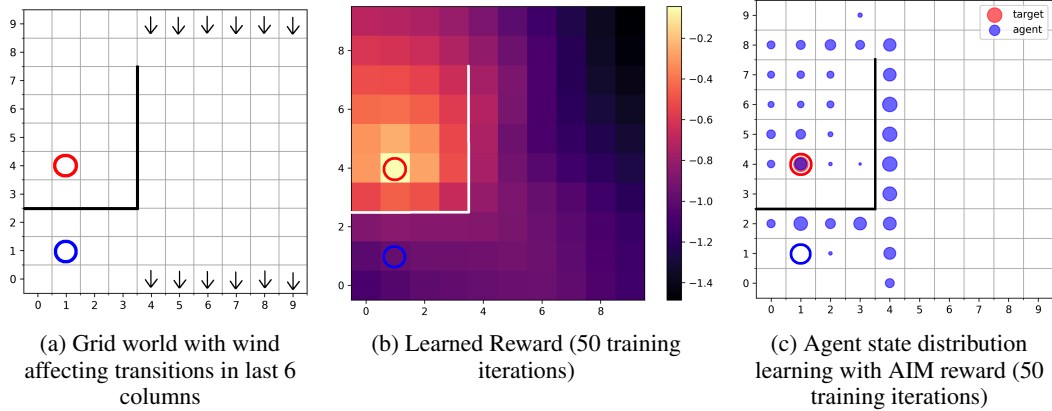

(a) Grid world with wind affecting transitions in last 6 columns

(b) Learned Reward (50 training iterations)

(c) Agent state distribution learning with AIM reward (50 training iterations)

Figure 4: Windy grid world (Figure 4a) experiments. The columns with arrows at the top and bottom have stochastic and asymmetric transitions induced by wind blowing from the top. Learned reward function (Figure 4b). Reward at each state of the grid world after training for 50 iterations with AIM. Hollow red circle indicates the goal state. White lines indicate the walls the agent cannot transition through. The agent's state visitation (Figure 4c): The hollow blue circle indicates agent's start state. The hollow red circle is the goal. Blue bubbles indicate relative time the agent's policy causes it to spend in respective states. Black lines indicate walls.

**Additional experiments**    We conducted variations form the basic experiment in the grid world to show that AIM and its novel regularization can learn a reward function which guides the agent to the goal even in the presence of stochastic transitions as well as transitions where the state features vary wildly from one step to the next.

First, we evaluate AIM's ability to learn in the presence of stochastic and asymmetric transitions in a windy version (Figure 4a) of the above grid world. Transitions in the last six columns of the grid are affected by a wind blowing from the top. Actions that try to move upwards only succeed $60\%$ of the time, and actions attempting to move sideways cause a transition diagonally downwards $40\%$ of the time. Movements downwards are unaffected. The rest of the experiment is carried out in the same way as above, but with $128$ hidden units in the hidden layer of the agent's Q function approximator (the reward function architecture is unchanged from the previous experiment). In Figure 4 we see that AIM learns a reward function that is still useful and interpretable, and leads to a policy that can confidently reach the goal, regardless of these stochastic and asymmetric transitions. Notice the effect of the stochastic transitions in the increased visitation in the sub-optimal states in the bottom two rows of column number $4$.

The next experiment tests what happens when the transition function causes the agent to jump between states where the state features vary sharply. As an example consider a toroidal grid world, where if an agent steps off one side of the grid it is transported to the other side. The distance function here should be smooth across such transitions, but might be hampered by the sharp change in input features. In Figure 5 we see show the policy and reward for a $10 \times 10$ toroidal grid world with start state at $(2, 2)$ and goal at $(7, 7)$. Transitions are deterministic but wrap around the edges of the grid as described above: a **down** action in row $0$ will transport the agent to the same column but row $9$. The start and the goal state are set up so that there are multiple optimal paths to the goal. The entropy maximizing soft Q-learning algorithm should take these paths with almost equal probability. From Figure 5 it is evident that AIM learns a reward function that is smooth across the actual transitions in the environment and allows the agent to learn a Q-function that places near equal mass on multiple trajectories.

Finally, we compare learning with AIM to the baselines mentioned in Section 6. RND, SMiRL, and MC were implemented and debugged on the grid world domain with a goal that is easier to reach before being used on the Fetch robot tasks. Hyper-parameters for the algorithms in both domains were determined through sweeps. In the Fetch domains, the hyperparameters for all three new baselines were decided on through sweeps on the FetchReach task, similar to how they were evaluated for AIM and the other baselines.

Figure 6 shows the results of executing these additional baselines on the grid world domain we use to motivate AIM. All the plots are taken after the techniques have had the same number of training iterations. However none of the baselines reach the goal even after providing additional time. We show the negative L2 distance to goal as a reward in the grid world domain to highlight that the DiscoRL [50] objective should not be considered equivalent to an oracle of the distance to goal. Note that RND (Figure 6c) explores most of the larger room early on, and then converges to the state distribution seen in the figure when it does not encounter the task reward. The SMiRL reward encourages the agent to minimize surprise, and the policy trained with this reward keeps the agent in the bottom left near its start state (Figure 6d).

## G  Statistical Analysis of the Results on Fetch Robot Tasks

To compare the performance of each method with statistical rigor, we used a repeated measures ANOVA design for binary observation where an observation is successful if an agent reaches the goal within an episode. We then conducted a Tukey test to compare the effects of each method, i.e., the estimated odds of reaching the goal given the algorithm. The goal of the statistical analysis presented here is twofold

1. Separate the uncertainty on the performance of each method from the variation due to random seeds.
2. Adjust the probability of making a false discovery due to multiple comparisons. This extra step is necessary to avoid detecting a large fraction of falsely "significant" differences since typical tests are designed to control the error rate of only one experiment.

The data for statistical analysis comes from $N_{\text{episodes}} = 100$ evaluation episodes per each one of $N_{\text{seeds}} = 6$ seeds. For all environments but FetchReach, these data is collected after 1 million environment interactions; and for FetchReach it is taken after 2000 interactions.

The repeated measures ANOVA design is formulated as a mixed effects generalized linear model and fitted separately for each one of the four environments

$$y_{ijk} \overset{\text{iid}}{\sim} \text{Bernoulli}(p_{ij}), \qquad\qquad k \in \{1, \ldots, N_{\text{episodes}}\}$$
$$\text{logit}(p_{ij}) = r_{\text{seed}_i} + \beta_{\text{algorithm}_j} \qquad i \in \{1, \ldots, N_{\text{seeds}}\}, j \in \{1, \ldots, N_{\text{algorithms}}\}$$
$$r_{\text{seed}_i} \overset{\text{iid}}{\sim} \text{Normal}(0, \sigma^2)$$

The variation due to the seed effects is measured by $\sigma^2$, whereas the uncertainty about the odds of reaching the goal using each algorithm is measured by the standard errors of the coefficients $\beta_{\text{algorithm}_j}$.

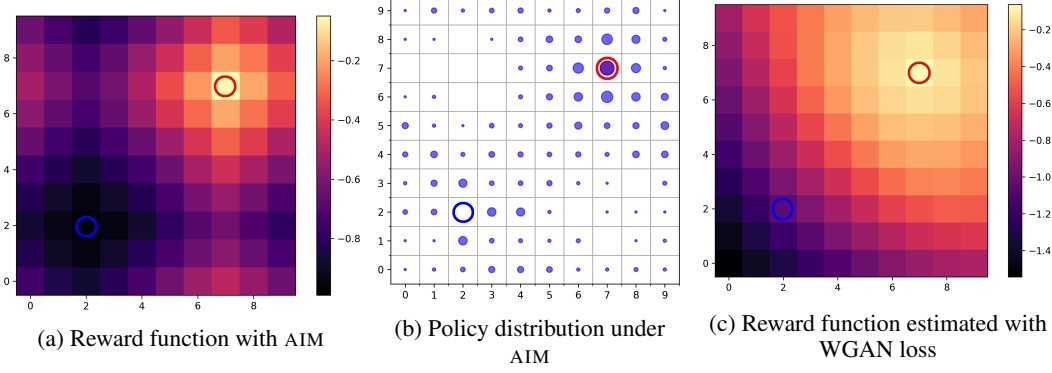

(a) Reward function with AIM

(b) Policy distribution under AIM

(c) Reward function estimated with WGAN loss

Figure 5: The reward function (Figure 5a) learned with AIM and subsequent policy distribution (Figure 5b) in a toroidal grid world, where an agent can transition from one edge of the grid across to the other. The hollow blue circle denotes the start state and the hollow red circle is the goal state. The reward function respects the sharp transitions from one end of the grid to the other. Conversely, if the reward function is learned using the WGAN objective [31] (Figure 5c), it does not respect the environment dynamics.

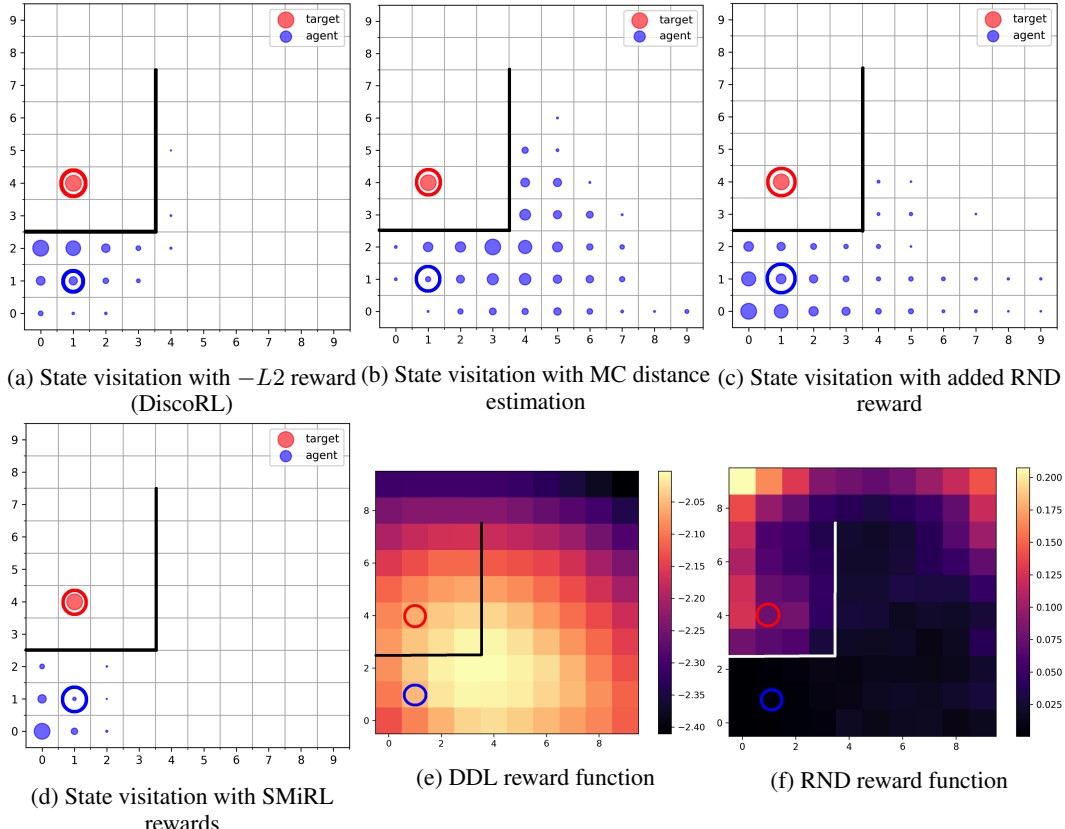

Figure 6: The state of the state visitation and reward functions for the new baselines. For camparison, Figure 2a shows the state visitation of policy trained using AIM. All algorithms are compared after 100 training iterations.

The Tukey test evaluates all null hypotheses $H_0$: $\beta_{\text{algorithm}_j} = \beta_{\text{algorithm}_{j'}}$ for all combinations of $j, j'$. To adjust for multiple comparisons each Tukey tests uses the Holm method. Since we are also doing a Tukey test for each environment, we further apply a Bonferroni adjustment with a factor of four. These types of adjustments are fairly common for dealing with multiple comparison in the literature of experimental design; the interested reader may consult [49].

The results, shown in Table 1, signal strong statistical evidence of the improvements from using the AIM learned rewards. In three of the four environments AIM and AIM+ R have similar odds of reaching the goal as the dense shaped reward ($H_0$ is not rejected,) and in all four environments AIM and AIM+ R have statistically significant higher odds of reaching the goal than the sparse reward ($H_0$ is rejected and $\beta$ is higher.)

| Contrast | Slide | Push | PickAndPlace | Reach |
|---|---|---|---|---|
| $\beta_{\text{AIM+R}} - \beta_{\text{HER+dense}}$ | 0.34 (0.14) | -1.74 (0.77) | -0.10 (0.45) | *-3.43 (0.34) |
| $\beta_{\text{AIM}} - \beta_{\text{HER+dense}}$ | 0.21 (0.14) | -2.19 (0.75) | *-1.50 (0.37) | *-5.01 (0.35) |
| $\beta_{\text{AIM+R}} - \beta_{\text{HER+sparse}}$ | *0.69 (0.13) | *5.32 (0.35) | *4.71 (0.33) | *4.75 (0.25) |
| $\beta_{\text{AIM}} - \beta_{\text{HER+sparse}}$ | *0.57 (0.13) | *4.86 (0.30) | *3.31 (0.19) | *3.17 (0.24) |

Table 1: Results of the Tukey test on the evaluation of Fetch tasks. The table entries are log odds ratios with standard deviations shown in parentheses. Positive values mean that AIM or AIM+R perform better than the method with negative sign in the contrast and viceversa. Asterisks mark statistical significance at 95%. If there is no asterisk, then $H_0$ is not rejected in which case the differences could be due to random chance.

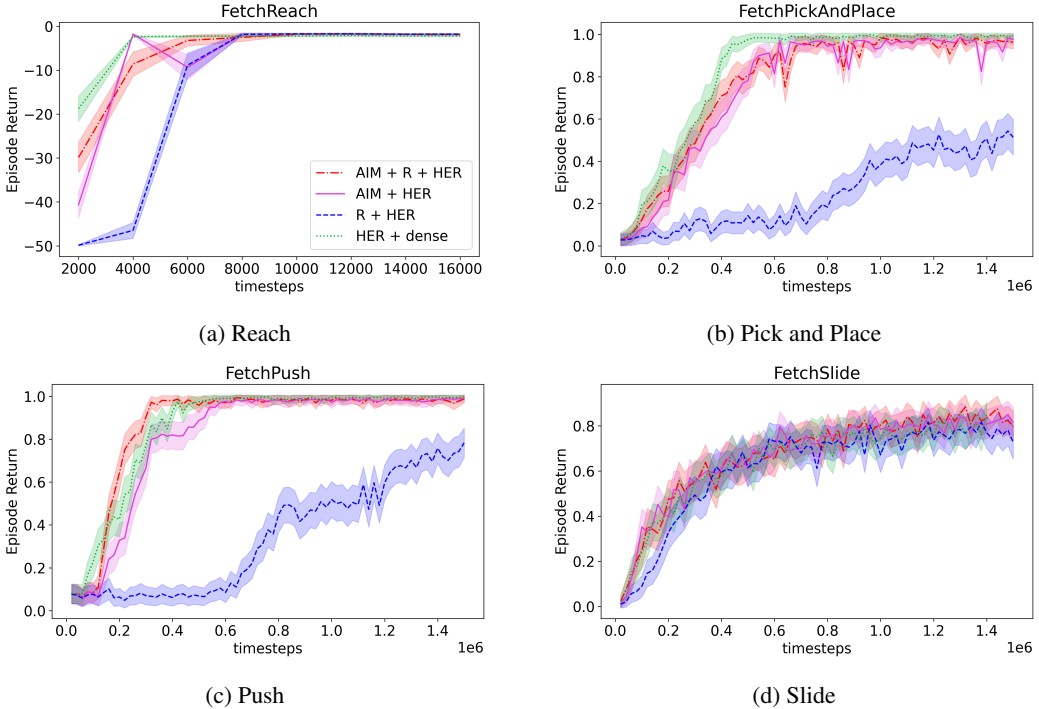

(a) Reach

(b) Pick and Place

(c) Push

(d) Slide

Figure 7: Comparing [AIM + HER] with an additional baseline which also uses the external task reward [AIM + R + HER]. The additional grounding provided by the external task reward allows the agent's learning to accelerate even further.

## H    Details of Experiments on Fetch Robot

The Fetch robot domain in OpenAI gym has four tasks available for testing. They are named Reach, Push, Slide, and Pick And Place. The Reach task is the simplest, with the goal being the 3-d coordinates where the end effector of the robot arm must be moved to. The Push task requires pushing an object from its current position on the table to the given target position somewhere else on the table. Slide is similar to Push, except the coefficient of friction on the table is reduced (causing pushed objects to slide) and the potential targets are over a larger area, meaning that the robot needs to learn to hit objects towards the goal with the right amount of force. Finally, Pick And Place is the task where the robot actuates it's gripper, picks up an object from its current position on the table and moves it through space to a given target position that could be at some height above the table. The goal space for the final three tasks are the required position of the object, and the goal the current state represents is the current position of that object.

Next, we note the hyperparameters used for various baselines as well as our implementation. The names of the hyperparameters are as specified in the stable baselines repository and used in the RL Zoo [59] codebase which we use for running experiments. Both the stable baselines repository and RL Zoo are available under the MIT license. These experiments were run on a compute cluster with each experiment assigned an Nvidia Titan V GPU, a single CPU and 12 GB of RAM. Each run of the TD3 baseline HER + R or HER + dense required 18 hours to execute, and each run which included AIM required 24 hours to complete execution.

TD3 [25], like its predecessor DDPG [47], suffers from the policy saturating to extremes of its parameterization. Hausknecht and Stone [35] have suggested various techniques to mitigate such saturation. We use a quadratic penalization for actions that exceed $80\%$ of the extreme value at either end, which is sufficient to not hurt learning and prevent saturation. Assuming the policy network predicts values between $-1$ and $1$ (as is the case when using the tanh activation function),

the regularization loss is:

$$L_a = \frac{1}{N}\sum_{i=1}^{N}[max(|\pi_\theta(s_i)|-0.8,0)]^2$$

where $N$ is the mini-batch size and $s_i$ is the state for the $i^{\text{th}}$ transition in the batch.

The other modification made to the stable baselines code is to use the Huber loss instead of the squared loss for Q-learning.

For evaluation, in the Reach domain the agent policy is evaluated for 100 episodes every 2000 steps. For the other three domains, the experiment is run for 1 million timesteps, and evaluated at every 20,000 steps for 100 episodes.

## H.1 TD3 and HER (R + HER)

| Hyperparameter | Value |
|---|---|
| n_sampled_goal | 4 |
| goal_selection_strategy | future |
| buffer_size | $10^6$ |
| batch_size | 256 |
| $\gamma$ (discount factor) | 0.95 |
| random_exploration | 0.3 |
| target_policy_noise | 0.2 |
| learning_rate | $1^{-3}$ |
| noise_type | normal |
| noise_std | 0.2 |
| MLP size of agent policy and Q function | $[256, 256, 256]$ |
| learning_starts | 1000 |
| train_freq | 10 |
| gradient_steps | 10 |
| $\tau$ (target policy update rate) | 0.05 |

## H.2 Dense reward TD3 and HER (dense + HER)

| Hyperparameter | Value |
|---|---|
| n_sampled_goal | 4 |
| goal_selection_strategy | future |
| buffer_size | $10^6$ |
| batch_size | 256 |
| $\gamma$ (discount factor) | 0.95 |
| random_exploration | 0.3 |
| target_policy_noise | 0.2 |
| learning_rate | $1^{-3}$ |
| noise_type | normal |
| noise_std | 0.2 |
| MLP size of agent policy and Q function | $[256, 256, 256]$ |
| learning_starts | 1000 |
| train_freq | 100 |
| gradient_steps | 200 |
| policy_delay | 5 |
| $\tau$ (target policy update rate) | 0.05 |

### H.3 TD3 and HER with AIM (AIM + HER) and (AIM + R + HER)

| Hyperparameter | Value |
|---|---|
| n_sampled_goal | 4 |
| goal_selection_strategy | future |
| buffer_size | $10^6$ |
| batch_size | 256 |
| $\gamma$ (discount factor) | 0.9 |
| random_exploration | 0.3 |
| target_policy_noise | 0.2 |
| learning_rate | $1^{-3}$ |
| noise_type | normal |
| noise_std | 0.2 |
| MLP size of agent policy and Q function | $[256, 256, 256]$ |
| learning_starts | 1000 |
| train_freq | 100 |
| gradient_steps | 200 |
| disc_train_freq | 100 |
| disc_steps | 20 |
| $\tau$ (target policy update rate) | 0.1 |