# OpenReview forum: "Adversarial Intrinsic Motivation for Reinforcement Learning"
_NeurIPS.cc/2021/Conference — NeurIPS 2021 Poster_

### Official Review · Reviewer_vqmu · 2021-06-27

**Rating:** 7
**Confidence:** 4

**Summary:**

This paper proposes a method for goal-conditioned RL. The main idea is to learn a function that looks like a value-function and then use that value function as a reward function for RL. This method is motivated as minimizing a certain Wasserstein distance. Empirically, the proposed method outperforms goal-conditioned RL applied to sparse reward functions, and performs roughly on par with goal-conditioned RL with dense rewards. In effect, it seems like this learned value function can serve the role of the dense reward function.

**Limitations And Societal Impact:**

The paper discusses limitations and potential broader impacts in the concluding section. I found this section a bit vague, and think it could be shortened and made more concise.

**Main Review:**

**Originality**: The method proposed in this paper can be broken down into two main ideas: (1) a new method for learning a value-function-like term, and (2) a method for using that value function.

The first idea is (to the best of my knowledge) novel and fairly clever. The idea is to formulate goal-conditioned RL as minimizing a Wasserstein-1 distance between the state marginal distribution and some "target" distribution centered at the goal (sometimes a Dirac, sometimes a Gaussian). The Wasserstein-1 distance requires defining an underlying metric; the paper proposes to use the shortest path distance as a (quasi-)metric. Cleverly, the paper shows how the dual form of the Wasserstein allows one to estimate the Wasserstein distance without being about the compute the shortest path distance (which, in general, is as difficulty as solving the goal-conditioned RL problem).

The second idea (using a value-function like term) is not novel. While the paper does not claim this to be a contribution, I'd recommend citing some prior work that employs the same idea (e.g., see [Hartikainen '19, Tian '20] or other prior work on using terminal value functions).

**Quality**:

While I'm not an expert on the Wasserstein distance, the derivation of the proposed method seems correct to me. However, the resulting method for learning the value function (Eq. 8) is almost exactly the same as standard temporal difference learning, which is *already* used by standard goal-conditioned RL methods. Precisely, temporal difference learning with the -1/0 reward can equivalently be viewed as solving a constrained optimization problem where (1) the goal state has value 0, (2) the difference between the values of consecutive states is less than 1, and (3) the value of all states is minimized. I suspect that Eq. 8 is equivalent to some linear-program formulation of the RL problem (see the Puterman MDP book). In summary, the resulting method bears a very close resemblance to (1) learning a value function and (2) using that value function as a reward function. This connection is useful because it not only explains what the "adversarial intrinsic motivation function" is (a value function, up to an additive constant), but also offers a potentially better way to learn it (using Monte Carlo regression instead of TD learning). I would highly encourage the authors to compare against this baseline (both with a value function learned via MC regression as in Eq. 3 of [Hartikainen '19]) and via TD learning (similar to Eq. 8 of this paper)).

Notwithstanding the results of this experiment, the idea that TD learning is equivalent to minimizing a Wasserstein-1 distance seems pretty cool! I'm not sure how this can be made into a practically useful insight.

Some additional comments about the analysis:
* It could be worth noting that the proposed objective is actually quite similar to C-learning [Eysenbach '20]. While C-learning minimizes a KL divergence with the goal distribution, the proposed method minimizes a Wasserstein-1 distance. After mentioning this connection, the authors could bring up the point about KL divergences not being informative for distributions with disjoint suppoer (perhaps adding another sentence (e.g., similar to the W-GAN paper) to build intuition.
* While the proposed method is derived for discrete-state MDPs, it is later applied to continuous state MDPs. I'd recommend including a sentence to explain that these experiments do not match theory (e.g., "Our experiments on simulated robotics tasks highlight that the method continues to work, even in settings with continuous states and actions, where our theory does not apply.").
* The discussion of variance in Sec 5.1 is reminiscent of risk-sensitive control (e.g., [Mihatsch '02]). This raises the question: if risk-sensitivity (i.e., minimizing variance) is important, why don't we just use a standard risk-sensitive objective (e.g., $\min \log E[T]$)?

A few questions for the authors:
1. In the robotic control tasks, it seems like the dense reward (a negative L2 distance) is equivalent to using the log probability of the target density (a Gaussian) as the reward. How is this different from what DiscoRL [Nasiriany '20] does. (If so, I'd recommend highlighting this connection, as it makes the comparison with dense rewards seems less like an "oracle" experiment and more like a prior method.)
2. Can the Wasserstein distance (Eq. 2) be expressed in terms of the expected time to reach the goal, or the probability of reaching the goal (i.e., without the bounds discussed in Sec 5.1)? Such a connection would help me gain intuition for exactly what the proposed method is optimizing.
3. If the optimal transport plan/coupling (L155) where known, are the clear ways that one could use it? Would it correspond to the optimal policy?


**Clarity**:

Generally, the paper was clearly written. For me, the most confusing part was the discussion of the Wasserstein metric. I think the confusion was in introducing the notion that the Wasserstein metric requires defining another metric function. I was also confused about whether the choice of this metric function was a design decision or was motivated by theory. Please see the minor comments at the end of the review.

**Significance**:

The idea that temporal difference learning is (approximately?) minimizing a Wasserstein metric seems potentially impactful, even if it doesn't immediately provide better algorithms for goal-conditioned RL. I don't think the actual method proposed will be impactful because (1) it seems nearly the same as prior methods that use a value function as a reward function, (2) the connection between the function $f(s)$ and standard quantities in RL (e.g., value functions) is unclear (but this could be easily corrected), and (3) the proposed method barely outperforms a simple baseline that uses a negative L2 distance as a reward function.

Minor comments:
* L59 - L60 -- If the policy minimizing the expected number of steps, then wouldn't such a policy also be optimal for stochastic dynamics?
* L69 - L70 -- This is a great introduction sentence!
* L87 -- Perhaps also discuss [Hartikainen '19].
* L120, L121 -- I was initially a bit confused what the notation "$\rho: \Delta (S)$" and "$\sigma: \Delta (G)$" meant.
* L129 -- I'd recommend stating $r(\bar{s}) = 1$ in math, too.
* L135 -- The three expectations could be combined into a single expectation.
* L184 "Definition 1" -- I found this definition pretty confusing, and would recommend cutting it and directly defining $d(s, s_g)$ using the equation after L193.
* L193 -- This equation raises an obvious question: Why not just use $T$ as a reward function? This question is only answered much later. I would recommend answering it immediately after defining $T$.
* L164 - L197 -- None of this material is novel. I'd recommend moving it to a new paragraph in Section 3.1 and just citing [Kaelbling '93].
* Eq 3 -- This idea is really clever! I'd highly recommend adding a few more sentences (and perhaps an illustration) to explain it.
* L207 - L208 -- I found this sentence confusing.
* L212 -- The importance of Proposition 1 is a bit unclear. I'd recommend moving it to Sec 5 (which focuses on analysis) and providing examples of settings where the inequalty is and is not tight.
* Sec. 5.1 -- The importance of this section is a bit unclear to me. The result I was expecting was that minimizing the proposed Wasserstein metric corresponds to minimizing expected hitting time. Unless the analysis shows this, I'd recommend moving all of Sec. 5.1 to the appendix.
* L240 "Adversarial Intrinsic Motivation" -- I found the term "intrinsic motivation" a bit confusing because it's often used in settings without clear objectives (in contrast, this paper is optimizing a very well defined objective). I'd recommend renaming it something like "Learning a Value Function to Minimize the Wasserstein-1 Distance".
* L244 "even if the ground metric" -- This point seems really important! I'd recommend bringing it up (or at least foreshadowing it) when "T" is first defined in L193.
* Fig 3: Please make this figure colorblind friendly.

-----------------------------
**Review decision**: Overall, I really like the idea of viewing goal-conditioned RL as a distribution matching problem and the idea of using Wasserstein duality to learn a value function. The paper clearly shows that this perspective yields good results on goal-conditioned RL problems. My main concern with the paper is its relationship with prior work, both empirically and theoretically. Empirically, it seems like the proposed method does not perform as well as "HER + dense", which I understand to be an instance of prior work [Disco RL applied to a Gaussian target distribution]. I would also recommend comparing against a baseline that learns a value function (either via TD learning or MC regression, as explained above) and uses that value function as a reward function. Theoretically, I'd highly recommend examining and explaining the connection between the adversarial function $f$ and temporal difference learning. Given these concerns, I vote to reject the paper.

**Update after discussion**: The additional experiments and clarifications provided during the rebuttal period have addressed by concerns, and I have changed my vote to "accept".



**Time Spent Reviewing:**

2-3

---

> ### Author Response · Authors · 2021-08-10
> **Response to Review by Reviewer vqmu**
>
> We thank the reviewer for providing their expertise in reviewing our work, and for the detailed feedback and references they have provided. We are grateful to the reviewer for pointing out the confusion between the Wasserstein distance between distributions and the ground metric between states used to compute this distance, and will modify the background section to clarify the difference. Additionally, we would like to thank them for pointing out the possibly very interesting connection to the LP-based formulation of policy evaluation, which bears resemblance to the algorithm proposed as an instantiation of adversarial intrinsic motivation.
>
> __Overall response__: The connection to the LP-based formulation of policy evaluation is surprising and potentially impactful, warranting further investigation (and thus, will be mentioned in a revised version). However,the objective of the potential function is not equivalent to this LP-formulation, which we address in detail below. Thus, we do not focus on such a connection in  this work (and do not include it in the title of the paper). We also emphasize that our theoretical analysis establishes a more general link between goal-conditioned RL and a type of Wasserstein distance minimization (based on the time step cost) that does not rely on the particular algorithmic instantiation of AIM that bears some resemblance to the LP formulation.
>
> Additionally, it is not the case that the dense reward baseline in our experiments is simply the reward function from the DiscoRL paper (the log probability of a standard gaussian centered at the goal). The oracle reward is equivalent to this reward function only when dealing with problems where the true distance between states is Euclidean (as is the case in these experiments). If the true distance to the goal were different (like the grid world domain, for example) then the DiscoRL reward would not coincide with the oracle reward. Specifically, these experiments in the Fetch robot domain (where we know the ground truth distance function) show that our approach is able to learn a reward equivalent to using the ground truth distance just from samples of the environment and simultaneously use it to learn the policy without needing significantly more samples, and without assuming that the true distance between states is Euclidean.
>
> We provide further clarifications below and detailed responses to R3’s comments:
>
> - __(Eq. 8) is almost exactly the same as standard temporal difference learning, which is already used by standard goal-conditioned RL methods. Precisely, temporal difference learning with the -1/0 reward can equivalently be viewed as solving a constrained optimization problem where (1) the goal state has value 0, (2) the difference between the values of consecutive states is less than 1, and (3) the value of all states is minimized. I suspect that Eq. 8 is equivalent to some linear-program formulation of the RL problem (see the Puterman MDP book). In summary, the resulting method bears a very close resemblance to (1) learning a value function and (2) using that value function as a reward function__:
> This characterization is imprecise, as we detail below. The LP formulation of policy evaluation in Puterman’s book (Section 6.9.1) is an objective that minimizes the value function on states the agent visits while enforcing the TD constraint. In contrast, our adversarial objective also maximizes the potential of the goal state (differentiating it from other states) even if the agent has not experienced it yet, unlike policy evaluation objectives. This crucial difference has a number of consequences.
> For one, the potential function during learning is not equivalent to the value of the agent’s policy. We verified this by attempting to use the potential function as the critic during learning.
> And second, a -1 reward at each time step induces the agent to explore, but it does not direct that exploration. In AIM, increasing the potential of the goal state encourages the agent to explore in a particular direction (namely, the direction of sharpest increase in potential), and thus it acts like the optimistic heuristic in the A* algorithm (as pointed out in our Discussion section). An illustration of this behavior is shown in the [HER + R] baseline in the Fetch Reach problem (Fig. 3) which already trains with a -1 reward per time step (similar to the suggested alternative for learning the reward function), and the AIM-learned reward helps the agent learn faster there as well.
>
>
> - __I'd recommend citing some prior work that employs the same idea (e.g., see [Hartikainen '19, Tian '20] or other prior work on using terminal value functions)__:
> We thank the reviewer for pointing out these works, and we will include them in our revision. We would like to point out that we have cited other work that uses a value function for shaped rewards [47], as well as work that uses trajectories in the environment to estimate the distance between states [68]. Our work is different from this latter one as well as the suggested papers in that it allows estimation of the potentials we use without actually experiencing a trajectory that reaches this goal, which we have not found any prior work capable of.
>
> - __It could be worth noting that the proposed objective is actually quite similar to C-learning [Eysenbach '20]. While C-learning minimizes a KL divergence with the goal distribution, the proposed method minimizes a Wasserstein-1 distance. After mentioning this connection, the authors could bring up the point about KL divergences not being informative for distributions with disjoint support (perhaps adding another sentence (e.g., similar to the W-GAN paper) to build intuition__:
> We thank the reviewer for noticing and pointing out this connection. We have already commented on C-learning [20] and its follow-up work [19] in our discussion of related work (Section 2.2), along with the comparison that the Wasserstein distance should be more informative when dealing with disjoint distributions. We will attempt to highlight this connection in additional discussion if possible.
>
> __Response to questions from the reviewer__:
> - __In the robotic control tasks, it seems like the dense reward (a negative L2 distance) is equivalent to using the log probability of the target density (a Gaussian) as the reward. How is this different from what DiscoRL [Nasiriany '20] does. (If so, I'd recommend highlighting this connection, as it makes the comparison with dense rewards seems less like an "oracle" experiment and more like a prior method.)__:
> Addressed in overall response.
>
> - __Can the Wasserstein distance (Eq. 2) be expressed in terms of the expected time to reach the goal, or the probability of reaching the goal (i.e., without the bounds discussed in Sec 5.1)? Such a connection would help me gain intuition for exactly what the proposed method is optimizing.__:
> Eq. 6 in section 5.1 already does something similar to what the reviewer suggests by expressing the Wasserstein distance in terms of both the expected hitting time and the Jensen gap, which is too a function of the hitting time. The purpose of the bounds in 5.1 is to explain that this Jensen gap is tied to the variance of the hitting time. Note that for an undiscounted finite-horizon MDP---not discussed in the paper since we focus on the discounted case---a simple expression for the Wasserstein distance (Eq. 5) can be derived as $\frac{1}{2}E[T + T^2]$ where $T$ is the hitting time. This expression makes it evident that the distance is minimized with smaller hitting times. The dependence on the variance is due to the quadratic term.
>
> - __If the optimal transport plan/coupling (L155) were known, are the clear ways that one could use it? Would it correspond to the optimal policy?__:
> To answer the second part of the question first, no, the optimal coupling does not tell us what the optimal policy would be. The optimal transport coupling tells us for each unit of measure in the first distribution, which unit of measure in the target distribution it should be matched to. It does not map out the path that the unit should take when being transported (which can be likened to the policy). Section B of the supplemental material extends the information on optimal transport and Wasserstein distance.
>
> __Response to the minor comments__:
> - __L59 - L60 -- If the policy minimizing the expected number of steps, then wouldn't such a policy also be optimal for stochastic dynamics?__:
> This is a subtle point, which we have tried to address through our analysis in Sections 4 and 5.1. To summarize briefly here, the optimal policy under discounted returns would prefer paths that have lower expected number of steps to the goal as well as a higher variance in these number of steps, whereas the policy that minimizes the Wasserstein distance would minimize the expected number of steps to the goal and prefer lower variance. When the environment dynamics are deterministic, the variance in the returns for the optimal policy goes to zero (since the optimal policy is deterministic), leading to our theorem.

---

> > ### Author Response · Authors · 2021-08-17
> > **Additional Results**
> >
> > As requested by the reviewers, we have run additional experiments to compare the performance of AIM with suggested baselines.
> > Those results can be accessed through this [anonymized link](https://drive.google.com/file/d/1tDRRB2SA4h3junIpPNWCgSx7-Be1oKJp/view?usp=sharing).
> >
> > We will incorporate these results into the main experiment section of our paper.
> >
> > We would like to highlight two baselines we compare to, which we had alluded to in our response above. First, we had hypothesized that learning the distance function via MC regression (similar to Hartikainen et al.) would not be equivalent to AIM. We validate our hypothesis by implementing MC regression of the distance between states reached during execution of the agent's policy, and then using the negative of distance estimate as the reward (which we refer to as [MC + R +HER]). We see that while this distance learning helps in some cases, it does not outperform AIM and in fact does not outperform the vanilla [R + HER] baseline across all the domains we test on. It also struggles to accurately estimate distances to goals it has not seen, as can be verified from the learned distance function in the grid world domain.
> >
> > Second, we show that the negative L2 distance (the reward equivalent to DiscoRL, Nasiriany et al.) cannot be considered as an oracle reward in all scenarios. If we use this reward in the grid world domain, the agent stays near its start state as it cannot go through the wall that the reward directs it towards.
> >
> > Additionally, we will also ensure that all the figures in the paper will be color-blind friendly. We have attempted to do so in the Fetch robot experiments linked above.

---

> > > ### Comment · Reviewer_vqmu · 2021-08-17
> > > **Response**
> > >
> > > Thank you for running these additional experiments!
> > >
> > > Quick question: I couldn't find the "negative L2 distance" results in the new results in Fig 7. If these results are added (and labeled as "negative L2 distance", or something like that), I will very likely change my score to "accept."

---

> > > > ### Author Response · Authors · 2021-08-17
> > > > **Response**
> > > >
> > > > Apologies for not being clear above.
> > > >
> > > > The "dense" reward we use in the Fetch domain is the negative L2 distance. In this domain, it can be considered the oracle distance. However, in our grid world domain we show that the negative L2 distance cannot always be considered the oracle distance (Figure 6 (a)). For clarity, we can relabel the dense reward and negative L2, and clarify our claim of it being the oracle reward.
> > > >
> > > > These experiments highlight that AIM is able to learn a reward function equivalent to using the oracle reward, and not one equivalent to using the DiscoRL reward.

---

> > > > > ### Comment · Reviewer_vqmu · 2021-08-21
> > > > > **Response**
> > > > >
> > > > > Thanks for clarifying that. With this additional clarification in place, I have changed score to accept.

---

### Official Review · Reviewer_wjsZ · 2021-07-14

**Rating:** 6
**Confidence:** 4

**Summary:**

This paper proposes a method that uses the Wasserstein-1 distance between policy State distributions in order to provide additional reward and intrinsic motivation for learning policies. The method comes up with an approximation for being able to compute and use the Wasserstein one distance that can work in some practical settings. The method in the paper is used to improve the learning performance on a collection of 2d navitaion tasks and fetching tasks.

Pros
The paper does propose an interesting way to be able to efficiently estimate the Wasserstein-1 distance that can be used to compute policies state visitation distributions distance.
The method is shown to accelerate learning on some example environments.

Cons
It's not entirely clear how to place this paper. The paper is proposing an intrinsic motivation method but the actual objective or method in the paper is a better way to estimate the Wasserstein-1 distance and how it can be used to match Target distributions as in a type of imitation learning. So then either the paper should have better comparisons and experiments that compare to other imitation learning methods that are also used to learn distance based metrics. Or if the paper is thoroughly proposing an intrinsic motivation method then the paper should discuss and compare to recent state-of-the-art intrinsic motivation methods like ICM or RND or even SMiRL.




Reference:
[1] D. Pathak, P. Agrawal, A. A. Efros, and T. Darrell.   Curiosity-driven Exploration by Self-498supervised Prediction. 2017
[2] Yuri Burda, Harrison Edwards, Amos Storkey, and Oleg Klimov. Exploration by random networkdistillation.ICLR, 2018b
[3] G. Berseth,  D. Geng,  C. M. Devin,  N. Rhinehart,  C. Finn,  D. Jayaraman,  and S. Levine.377{SM}irl: Surprise minimizing reinforcement learning in unstable environments. InInternational378Conference on Learning Representations, 2021.


**Main Review:**



Additional comments on the paper:
- The work in this paper sounds very familiar to the state space covering paper [4] that also trained an objective function in order to have an agent minimize the distance between a type of distribution but in that case, the distribution was covering the entire State space.
- Section 2.1 is outlining previous methods and intrinsic motivation and ends with a sentence saying that the Wasserstein distance has been used before as a valid reward for imitation learning. This is making it sound like the use of this metric that's proposed in the paper is already been done in two previous works. It would be helpful if the authors could clarify these points by noting the differences between the new proposed method and those previous works.
- As the authors note, the Wasserstein distance may be more informative when distributions are disjoint, theoretically. It will be very important to be able to show in the paper that in practice the method proposed can perform much better than previous works [4] in this area that's solving a very similar problem. It would also be helpful to back up the claim online 97 stating that AIM might induce a form of directed exploration.
- In terms of the focus on the type of problems in the paper, it would be very helpful to understand the reasoning why problems on unrealizable target distributions that cannot be completely matched. Is this a common challenge in reinforcement learning that needs to be addressed and if it is addressed will assist in solving a number of problems that the community is interested in?
- The experiments in the paper describe how aim May speed up learning a policy towards a single goal. This makes one wonder whether or not the method in the paper can also speed up learning towards multiple goals and whether or not the single goal chosen has any particular biased towards the way AIM works. Were multiple goals evaluated in each of the environments in the experiments?
- It's not completely clear how figure 2 is showing that AIM enables the agent to reach the goal and learn the required policy quickly. To understand if it's learning a policy quickly it will need to be compared to some other method that is also plotting these types of details and state visitation distribution. It is hard to tell if the visualization for figure two really indicates how or what the agent has learned and what it should follow in order to help it get to the Target better. It almost seems like figure two a to a has learned a better State visitation distribution that will lead it more quickly to the goal?
- The results in figure 3 appear to be rather similar. Most of the methods seem to have very similar performance with the exception of reward plus HER. It would be beneficial to have results on additional environments to help readers understand what type of environments AIM will work best on. There are also comparisons to other aiAIM like methods that include RND [2] that would be helpful to compare to to make sure AIM is an improvement over the current state of the art.

If these issues area addressed I would consider updating my score.


[4] Lee, L., Eysenbach, B., Parisotto, E., Xing, E., Levine, S., & Salakhutdinov, R. (2019). Efficient exploration via state marginal matching. arXiv preprint arXiv:1906.05274.

--------- Post author response -----
I have updated my score to reflect clarifications from the authors.

**Time Spent Reviewing:**

2.5

---

> ### Author Response · Authors · 2021-08-10
> **Response to Review by Reviewer wjsZ**
>
> We thank the reviewer for providing their expertise in reviewing our work, remarking that they found our ideas interesting, and the thoughtful feedback and recommendations provided.
>
> It appears that the reviewer may have some misconceptions that led to the difficulty in placing our work. We will endeavour to clarify this issue and address it briefly here. First, while we use an objective that could be considered similar to imitation learning, we use the state and target distributions in the context of goal-conditioned RL, and our target is not the distribution induced by any policy. As such, our approach cannot really be considered imitation learning. Second, while intrinsic motivation has recently been associated with techniques that enable better exploration (such as RND, ICM, and SMiRL), they have also been used more generally to refer to reward functions learned by the agent and used to enhance the learning of its policy or other skills (For example, [10, 60, 62, 63, 72, 73] from our draft). This is the context in which we refer to our approach as an intrinsic motivation technique.
>
> In response to the additional comments:
> - __Comparison to [4]__: The paper that the reviewer refers to is cited in our draft as [41], and is also similar to [33]. [33] and [41] both consider a state distribution matching problem with the target given as a uniform distribution over the state space. These techniques look to minimize the KL divergence to such a target distribution. In particular, the algorithm in [41] explicitly tries to estimate the agent's state visitation distribution and then computes the KL divergence samples for experienced states. In the goal-conditioned RL case, we expect the target goal distribution to be disjoint compared to the agent distribution, and thus this KL term will be infinite and uninformative. Experimentally, we have compared to FAIRL [25], which minimizes a forward KL divergence to a given target distribution, which is similar to these techniques.
> - __Comparison to related work that used negative Wasserstein distance as reward__: The Wasserstein distance has been used for imitation learning before in the papers pointed out in related work. In contrast, we show that minimizing the Wasserstein distance can also be useful for goal-conditioned RL, as seen from our analysis (Section 5.1). Further, these works rely on the L2 distance as the ground metric. We suggest the use of the time-step distance, which we argue is a more natural quasi-metric to use when dealing with sequential decision making problems.
> - __As the authors note, the Wasserstein distance may be more informative when distributions are disjoint, theoretically. It will be very important to be able to show in the paper that in practice the method proposed can perform much better than previous works [4] in this area that's solving a very similar problem. It would also be helpful to back up the claim online 97 stating that AIM might induce a form of directed exploration.__: As highlighted earlier, the technique used in [41] cannot be applied to goal-conditioned RL since the subsequent reward function would be uninformative. The failure of FAIRL on the robotics experiments seem to validate our hypothesis that these methods might not work well on such disjoint distributions. To showcase the directed exploration that we think AIM engenders, we would like to draw attention to the grid world experiment where the agent needs to get around two walls and backtrack to find the goal. Here, the agent did not need to explore the environment extensively to find the goal, instead focusing its efforts on trying to find paths to the goal. This directed exploration can be seen from Figure 2 (a), where the agent’s state distribution shows that it does not explore the right side of the grid. More details on Figure 2 further below.
> - __In terms of the focus on the type of problems in the paper, it would be very helpful to understand the reasoning ...__: Goal-conditioned RL problems are extremely common in traditional as well as modern RL, and our method would be immediately applicable and useful in solving them. Additionally, future work towards more complex target distributions and an analysis of the policies that minimize the Wasserstein distance to them could increase the applicability of AIM to other RL problems.
> - __multiple goals__: Extending the previous explanation to how our approach might work if considering multiple goals, we did evaluate our algorithm on multiple goals in the grid world environment as part of our preliminary experiments. We observed that the agent finds a path that goes through all the goals as efficiently as possible, if reaching one goal does not terminate the episode. More detailed experiments and analysis were deferred for future work, however, given the scope of this paper. This question is definitely an interesting and nuanced one that we plan to address in the future.
> - __Figure 2__: It is unfortunate that Figure 2 was not clear about what it was showing. We give more details here and will update the caption for clarity. Figures 2(a) and 2(b) are figures that showcase our method, while Figure 2(c) is the comparison with an agent which is learning with task rewards, but trained for 5 times longer. 2(a) is the resultant state visitation distribution of an agent that uses only rewards learned through AIM, and shows through its state visitation distribution that it has learned to reach the goal while not needing to explore the state space extensively to do so. Figure 2(b) shows the reward function that was learned, and provides intuition for how the agent is guided to the goal. With Figure 2 (c) we highlight with the state visitation distribution of a maximum entropy policy with task rewards that it is not enough to search out and reliably learn a policy for goals that are hard to reach.
> - __Figure 3__: Finally, the reviewer suggests that “The results in figure 3 appear to be rather similar.” We would like to clarify that in Figure 3, we compare two variants of our approach ([AIM + HER] and [AIM + R + HER]) to the task reward ([R + HER]) and an oracle reward that gives the distance to the goal ([dense + HER]). The latter is meant as an upper bound since it leverages additional information that is not available to the baselines or our approach. It shows that [AIM + HER] can learn the distance to goals and present a reward signal as informative as knowing the true distance to the goal, without needing too many extra samples. Adding the task reward [AIM + R + HER] does not improve performance substantially, showing that AIM learns a reward function that is informative enough on its own. Additionally, we attempted to compare to FAIRL, but note in the paper that it fails to reach the goal unless its reward function is modified.

---

> > ### Comment · Reviewer_wjsZ · 2021-08-11
> > **Reviewer response**
> >
> > I would like to thank the authors for their feedback. Especially their discussion on the positioning of the paper. I have updated my score to reflect the clarifications they provided. However, due to the clarification on the positioning of the paper the last comment on the experiments, additional comparisons, or additional environments is now more important. If the method is an intrinsic reward method it should be compared to other intrinsic reward methods on environments from those works. For example, comparing to [RND + HER + R], [ICM + HER + R] or [SMiRL + HER + R]. This analysis would increase my confidence in the state-of-the-art performance of the method and I will further increase my score.

---

> > > ### Author Response · Authors · 2021-08-17
> > > **Additional Results**
> > >
> > > We would like to thank the reviewer for considering our response.
> > > As requested, we have run additional experiments to compare the performance of AIM with some of the suggested baselines.
> > > Those results can be accessed through this [anonymized link](https://drive.google.com/file/d/1tDRRB2SA4h3junIpPNWCgSx7-Be1oKJp/view?usp=sharing).
> > >
> > > We will incorporate these results into the main experiment section of our paper. In particular, we have implemented the [RND + R + HER] baseline as requested. Its performance in our grid world environment as well as the Fetch domain when compared to AIM shows that the reward function learned through AIM presents more learning signal and directs the agent more clearly towards the goal. We hope the other baselines we compare to are also helpful in highlighting the value that AIM brings to this problem.

---

> > > > ### Comment · Reviewer_wjsZ · 2021-08-18
> > > > **Re: experiments**
> > > >
> > > > I do appreciate that the authors have included more baselines, including RND. However, I am still concerned that this analysis does not include any of the environments from the RND paper. The performance of these methods could be very different in Atari environments or anything that has image-based inputs. If these are completed I am likely to increase my score to a 7.
> > > >
> > > > It will also be important to describe how the new baselines were implemented to make sure the new results illustrate a fair comparison. For example, It is tricky to balance the intrinsic reward from RND and the environment reward. What have the authors done to help ensure there is an equal share of reward signal from RND and the environment? The hyperparameter sweeps are helpful but more details on the parameters for the new baselines are important to include.

---

> > > > > ### Author Response · Authors · 2021-08-18
> > > > > **Response to Re: experiments**
> > > > >
> > > > > Thank you for considering our additional experiments with some of the suggested baselines. With regards to evaluating on environments used in the RND paper, we would like to point out that the Atari domains have not been previously used for goal-conditioned RL and do not come with an accepted set of goals that we could evaluate our goal-conditioned policy on.
> > > > >
> > > > > The domains we tested on were chosen to showcase the failure of current goal-conditioned RL techniques, as a proof of concept of how our method addresses them. The fact that RND does not work well in these domains further highlights the dearth of general techniques applicable to the problem. There are many other domains where this method could apply as well, but the purpose of the experiments is to show examples of the needs for (and utility of) our method, not to provide an exhaustive enumeration of possible applications.
> > > > >
> > > > > With regards to how RND was implemented. We used MLPs with two hidden layers of the same width as used by our Wasserstein potential function (n=64), and the outputs of the MLP were also 64. The inputs to the RND networks were normalized with a running mean and standard deviation as specified in the paper, and the reward was normalized with a running standard deviation. The RND reward was added to the task reward with a weighting chosen through a hyperparameter sweep. We experimented with the following weights on the RND reward before adding it to the task reward: $[0.1, 0.3, 0.5,  1., 3.]$.
> > > > >
> > > > > In the grid world domain, we checked the RND reward with goals separately set at locations $[8, 1]$ (bottom right side of the grid) and $[6, 8]$ (top center-right of the grid) to ensure that the RND rewards were not overwhelming the task rewards and to check that they were ensuring exploration. The final weighting that worked in the grid world was $0.3$.
> > > > >
> > > > > For the Fetch domains, the final weight chosen was $1.$. This weight was first chosen through sweeps on the Fetch Reach task, and then validated by checking the weights $[0.5, 1., 1.5]$ on Fetch Push to ensure that we were not overfitting to the Reach task. We further validated this choice by checking that the mean of the RND reward was less than $0.1$ in the Fetch tasks, ensuring that it would not overwhelm the task reward.

---

> > > > > > ### Author Response · Authors · 2021-08-24
> > > > > > **Additional Response to Re: experiments**
> > > > > >
> > > > > > As the reviewer suggested earlier, we also evaluated whether the SMiRL reward bonus [1] could be of use for goal-conditioned RL. We tried several configurations, but found SMiRL to be counterproductive for this problem. Specifically, the surprise minimizing quality of SMiRL, in an environment where any stochasticity comes from the agent itself, encourages a policy that does not move from its start state. The additional results with SMiRL are at this [new link](https://drive.google.com/file/d/1KYeYkvRW9kTTYDweJ3xyabeYKTY_DhiM/view?usp=sharing). SMiRL results and analysis are added at the end.
> > > > > >
> > > > > > Further, the reviewer inquired about testing against RND in a domain where its strength has been validated (the Atari domain or other image based domains). While in principle AIM should work in such conditions, its application to image-based domains raises some challenges that go beyond the scope of this paper, and thus would be more appropriate to address in a separate future work.
> > > > > >
> > > > > > Specifically, there is currently no standard accepted method for applying goal-conditioned RL to visual domains. Evaluating goal-conditioned RL on visual inputs will thus require various non-trivial decisions regarding the framing of the problem. For example, what a goal should be --- whether it should be the full image (requiring that the goal is identifiable from irrelevant visual information) [2] , or just the relevant goal-specific information (such as the location of the object in the Fetch tasks)[5] --- and how it should be represented [2, 3, 4, 5]. Additionally, being able to set and extract goal states requires additional control over the environment for implementing techniques like HER. In addition, the suggested Atari domain, where RND was evaluated, is not a domain widely used for evaluating goal-conditioned policies. Thus evaluating in that setting would not lead to natural comparison points.
> > > > > >
> > > > > >
> > > > > > References:
> > > > > > - [1] Berseth et al., SMiRL: Surprise Minimizing Reinforcement Learning In Unstable Environments, ICLR 2021.
> > > > > > - [2] Nair et al., Visual Reinforcement Learning with Imagined Goals. Neurips 2018
> > > > > > - [3] Nair et al., Contextual Imagined Goals for Self-Supervised Robotic Learning, CoRL 2019
> > > > > > - [4] Chane-Sane et al., Goal-Conditioned Reinforcement Learning with Imagined Subgoals, ICML 2021
> > > > > > - [5] Warde-Farley et al., Unsupervised Control Through Non-parametric Discriminative Rewards, ICLR 2019

---

> > > > > > > ### Comment · Reviewer_wjsZ · 2021-08-26
> > > > > > > **Re: Additional Analysis**
> > > > > > >
> > > > > > > I appreciate the additional analysis the authors have provided for the paper. The contribution is limited to methods based on goal-conditioned learning, however, this is not unreasonable. I do agree that there is no standard method to apply goal-conditioned RL to Image-based tasks. While [2,3] might be a good place to start it is not necessary for this submission. I do look forward to future work that may combine methods similar to [2,3] with AIM.

---

### Official Review · Reviewer_Qdoa · 2021-07-16

**Rating:** 7
**Confidence:** 3

**Summary:**

This paper proposes Adversarial Intrinsic Motivation (AIM), a state distribution matching algorithm for goal-conditioned reinforcement learning. State distribution matching is an effective approach to imitation learning where the target distribution is yielded by the expert behavior. However, it is unclear how to apply it to goal-conditioned RL because the target distribution is unachievable - it puts all mass on the goal states and none on the trajectories leading to them. This paper proposes to use Wasserstein-1 distance as the distance measure between the agent's state visiting distribution and the goal distribution. To measure the distance between two states, the authors propose a quasimetric specific to MDPs - the time-step metric. The authors then prove that the policy that minimizes the Wasserstein-1 distance between the agent's state visiting distribution and the goal distribution under the time-step metric also minimizes the steps needed to achieve the goal. Empirical results show that AIM can provide dense and effective reward signals to the agent so that it learns better than using only the sparse reward of achieving the goal. When combined with Hindsight Experience Replay (HER), AIM outperforms the sparse goal-achieving rewards in a goal-conditioned RL setup and approaches the performance of a near-oracle hand-designed reward signal.

**Limitations And Societal Impact:**

The main contribution of this paper is theoretical and the experiments are mostly for illustration purposes. I do not see any potential negative societal impacts of this paper.

**Main Review:**

Overall this paper is well written and easy to follow. The idea of using Wasserstein-1 distance to measure the distance between the state visiting distribution and the unachievable goal distribution makes sense. The theoretical analysis provides insight into the solution found by minimizing the Wasserstein-1 distance. The experiments successfully demonstrate the merits of AIM. I especially like the thorough analysis in the experiment section where the authors provide detailed explanations to both the successful cases and the failed cases.

I do have one suggestion regarding the experiment. One natural baseline I can think of is to combine GAIL with HER in the goal-conditioned setup where the trajectories leading to the hindsight goals are treated as expert trajectories. So far I do not see an obvious reason why AIM would be better than this baseline. Thus I think this can be a strong baseline to compare against.

**Time Spent Reviewing:**

4

---

> ### Author Response · Authors · 2021-08-10
> **Response to Review by Reviewer Qdoa**
>
> We would like to thank the reviewer for providing their expertise in reviewing our work, their positive remark on the readability of the paper, and their feedback and recommendations. Our response here focuses on the conceptual difference between the suggested baseline (GAIL + HER on experienced trajectories) and AIM. We would like to point out that we have compared to FAIRL, which is an extension of GAIL that minimizes the Forward KL divergence and has been shown to cover some state distributions if the discriminator is regularized similar to WGAN [29], with the same target distribution we use for AIM. We maintain that this comparison (while not exactly what the reviewer has suggested) is close enough that our assertion that these techniques will not work well is supported.
>
> Even if we did exactly what the reviewer suggested, we hypothesize that it would not work as well as the reviewer suggests, for the following two reasons. First, GAIL aims to match a given distribution (while AIM tries to minimize the distance to an unachievable distribution). So if the hindsight demonstrations are sub-optimal, GAIL will learn to imitate these sub-optimal trajectories. Second, HER already helps the agent learn how to reach the goals it has experienced before, so GAIL might not add as much value as expected. We use HER with AIM for the additional learning data and stabilizing influence of reachable goals on the potential function. In fact, in the grid world experiment we do not use any hindsight and AIM still learns to reach the goal. It is unclear how GAIL could be used in such a setting.

---

> > ### Author Response · Authors · 2021-08-17
> > **Additional Experiments**
> >
> > As requested by the reviewers, we have run additional experiments to compare the performance of AIM with suggested baselines.
> > Those results can be accessed through this [anonymized link](https://drive.google.com/file/d/1tDRRB2SA4h3junIpPNWCgSx7-Be1oKJp/view?usp=sharing).
> >
> > We will incorporate these results into the main experiment section of our paper. In particular, we have implemented the [GAIL + R + HER] baseline as requested. As we hypothesized, it sometimes helps the agent to learn but does not do as well as AIM, since it does not help direct the agent for goals it has not seen yet. Thus [GAIL + R + HER] is not equivalent to AIM.

---

> > > ### Comment · Reviewer_Qdoa · 2021-08-31
> > > **Thank you for the response**
> > >
> > > Thank you for the response. The additional empirical results further strengthen the paper. I will stick to my initial evaluation and vote for acceptance.

---

### Decision · Program_Chairs · 2021-09-27

**Decision:**

Accept (Poster)

**Comment:**

This paper presents an elegant idea and a fine investigation -- an important contribution to the community. After clarifications made by the authors, all reviewers agreed that this paper should be accepted.
When preparing the final version, please make sure to address the reviewers' feedback.